# New investigations on homogeneous ice nucleation:
# the effects of water activity and water saturation formulations

Manuel Baumgartner[1,2,a], Christian Rolf[3], Jens-Uwe Grooß[3], Julia Schneider[4], Tobias Schorr[4], Ottmar Möhler[4], Peter Spichtinger[1], and Martina Krämer[1,3]

[1] Institute for Atmospheric Physics, Johannes Gutenberg University, Mainz, Germany
[2] Zentrum für Datenverarbeitung, Johannes Gutenberg University, Mainz, Germany
[3] Forschungszentrum Jülich GmbH, Institute of Energy and Climate Research 7 – Stratosphere, Jülich, Germany
[4] Institute of Meteorology and Climate Research, Karlsruhe Institute of Technology, Karlsruhe, Germany
[a] now at: Deutscher Wetterdienst, Offenbach, Germany

**Correspondence:** Manuel Baumgartner (manuel.baumgartner@dwd.de)

**Abstract.** Laboratory measurements at the AIDA cloud chamber and airborne in-situ observations suggest that the homogeneous freezing thresholds at low temperatures are possibly higher than expected from the so-called 'Koop-line'. This finding is of importance, because the ice onset relative humidity affects the cirrus cloud coverage and, at the very low temperatures of the tropical tropopause layer, together with the number of ice crystals also the transport of water vapor into the stratosphere.

5  Both, the appearance of cirrus clouds and the amount of stratospheric water feed back to the radiative budget of the atmosphere. In order to explore the enhanced ice onset humidities, we re-examine the entire homogeneous ice nucleation process, ice onset and nucleated crystal numbers, by means of a two-moment microphysics scheme embedded in the trajectory based model (CLaMS-Ice) as follows: the well-understood and described theoretical framework of homogeneous ice nucleation yet includes certain formulations of the water activity of the freezing aerosol particles and the saturation vapor pressure of water with respect to liquid water. However, different formulations are available for both parameters. Here, we present extensive sensitivity simulations testing the influence of three different formulations for the water activity and four for the water saturation on homogeneous ice nucleation. We found that the number of nucleated ice crystals is almost independent of these formulations but is instead sensitive to the size distribution of the freezing aerosol particles. The ice onset humidities, also depending on the particle size, are however significantly affected by the choices of the water activity and water saturation, in particular at cold temperatures $\lesssim 205\,\mathrm{K}$. From the CLaMS-Ice sensitivity simulations, we here provide combinations of water saturation and water activity formulations suitable to reproduce the new, enhanced freezing line.

## 1   Introduction

A detailed understanding of the formation processes of ice in the atmosphere remains one of the most challenging topics in cloud physics. Until now, two major formation pathways have been identified at cold temperatures below about $235\,\mathrm{K}$: homogeneous and heterogeneous nucleation of ice (see, e.g., Vali et al., 2015, for the terminology). Heterogeneous ice nucleation is defined as the occurrence of ice crystals, where another substance (e.g. mineral dust or soot) aids to initiate the freezing process.

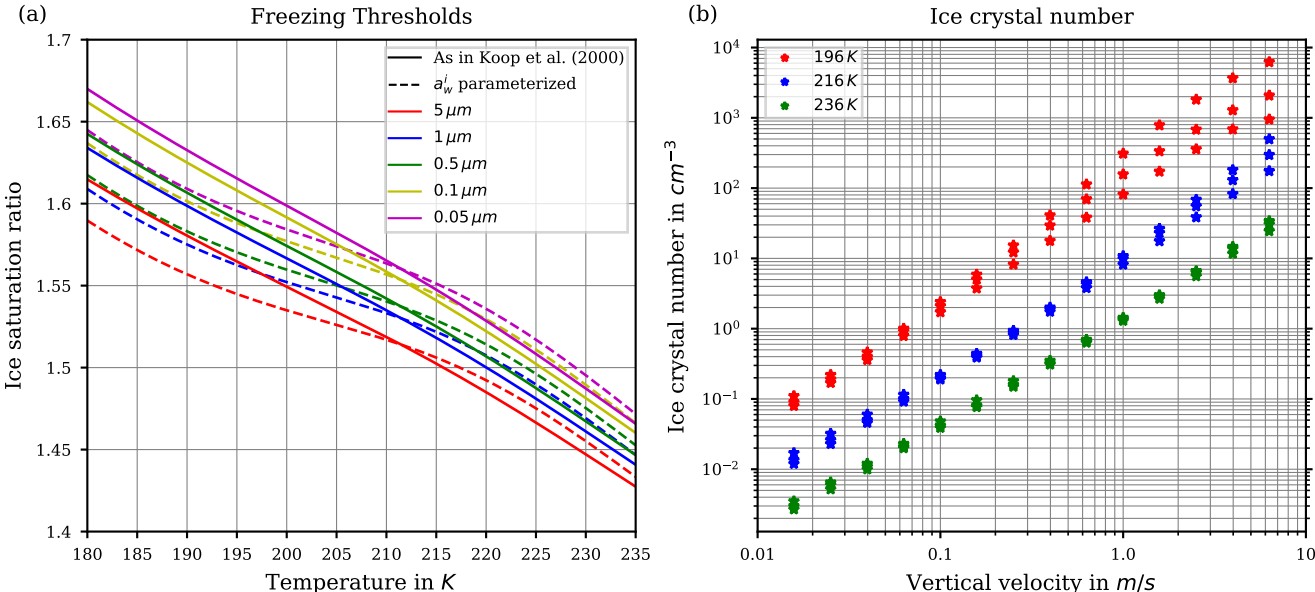

**Figure 1.** Panel a: Freezing thresholds as computed in Koop et al. (2000, reproducing their figure 3) as a function of temperature for various radii of the solution particles. The solid lines are based on the formulation $a_w^i(T) = \frac{p_{i,\text{sat}}(T)}{p_{\text{sat,MK}}(T)}$ for the water activity of ice $a_w^i$, where $p_{i,\text{sat}}$ is the saturation vapor pressure of ice and $p_{\text{sat,MK}}$ is the saturation vapor pressure of liquid water, both according to the formulations given in Murphy and Koop (2005). The dashed lines use the formulation of $a_w^i$ as given in Koop et al. (2000, their equations 1 and 2) and are converted into a saturation ratio with the help of the formulations $p_{i,\text{sat}}$ and $p_{\text{sat},MK}$ from Murphy and Koop (2005). Panel b: Number concentration of homogeneously nucleated ice crystals as given in Kärcher and Lohmann (2002, reproducing their figure 3).

As may be anticipated, understanding of heterogeneous ice nucleation is difficult due to the numerous possible particle types that potentially participate in the freezing process (Hoose and Möhler, 2012). In contrast, homogeneous nucleation refers to the spontaneous freezing of pre-existing solution particles and is comparatively well understood, although not on a molecular level but in the sense of its bulk-behaviour. Note, in this study we consider the homogeneous freezing of small aqueous aerosol particles, a process that is important for the formation of in-situ origin cirrus clouds. Formation of liquid origin cirrus clouds is governed by the freezing of much larger cloud droplets which typically freeze heterogeneously already at warmer temperatures.

Two decades ago, Koop et al. (2000) introduced, based on laboratory bulk experiments, a new, relatively simple theory describing homogeneous ice formation, whose details are given in Section 2. From this theory, hereafter also called "Koop's approach", the supersaturation with respect to ice at which ice is formed (the so-called *freezing threshold*) can be derived as a function of temperature, see Section 2. The freezing threshold is shown in Figure 1a (solid lines) and is a function of temperature and radius of the pre-existing solution particles. It may be interpreted as a probability of freezing of a solution particle. This theory is nowadays called the 'Koop parameterization' and is adopted in nearly every large-scale numerical model to reproduce the homogeneous ice formation process.

However, in freezing experiments at the large cloud chamber AIDA (Schneider et al., 2021), higher freezing thresholds for binary $H_2SO_4/H_2O$ solution particles were observed than predicted by the aforementioned freezing threshold derived from Koop et al. (2000). This is particularly true for experiments at cold temperatures, e.g. below about $205\,\mathrm{K}$, and provides a major motivation for the current study. Higher freezing thresholds at cold temperatures translate into a delayed freezing of the solution particles within a vertically ascending air parcel in the upper troposphere. Such a delay impacts our understanding

of the freeze drying at the tropical cold point tropopause. More precisely, a delayed cirrus cloud formation implies a reduced cirrus cloud cover, thus also higher values of the ice saturation ratio, being consistent with actual observations (Krämer et al., 2009, 2016, 2020; Jensen et al., 2005; Lawson et al., 2008). Prevailing larger values of the ice saturation ratio additionally imply an increased amount of water vapor that can be transported across the tropopause into the stratosphere (Rollins et al., 2016; Krämer et al., 2020), which directly affects the climate (e.g., Solomon et al., 2010; Riese et al., 2012).

Apart from the physical understanding of the ice formation processes, another important aspect is their representation in (numerical) models. Numerical cloud models typically implement a bulk microphysics scheme by predicting the mass concentration of the different hydrometeor types (one moment schemes) or additionally their number concentration (two moment schemes). Such bulk-schemes may be thought of as predicting mean values of the quantities of interest, i.e. at each timestep only the total mass and/or the total number of the involved hydrometeor types are computed. Although there also exist so-called

bin microphysics schemes which predict the evolution of the size distributions of the hydrometeor types, bulk microphysics schemes are preferred in large-scale numerical models due to their considerable lower computational demands (an overview is provided in Khain et al., 2015, 2000).

An approach to investigate the similarities and differences between microphysical schemes is to compare them in idealized settings, e.g. by prescribing a simple flow pattern (e.g., Morrison and Grabowski, 2007) or to compute the temporal evolution

of the microphysics scheme along a prescribed air parcel trajectory. The latter approach has the advantage of simplicity, i.e. not only numerous numerical experiments are possible but also theoretical investigations (Baumgartner and Spichtinger, 2019).

Irrespective of the chosen microphysics scheme, several physical quantities need to be computed, which is usually done by evaluating parameterizations for these quantities. One particular example is the saturation vapor pressure of liquid water. There exist numerous different parameterizations to approximate the true value at a given temperature (a review is provided

in Murphy and Koop, 2005), but particularly at cold temperatures, their values differ significantly. Emerging questions in this context include:

1. How does the choice of a parameterization affect the computed results?

2. How do the computed results compare to laboratory observations, e.g. observations made at the AIDA cloud chamber (Schneider et al., 2021)?

3. How do the computed homogeneously nucleated ice crystal numbers compare to the reference results from Kärcher and Lohmann (2002)?

The last question addresses the fact, that Kärcher and Lohmann (2002) provided reference computations and a parameterization for the number of homogeneously nucleated ice crystals as a function of temperature and updraft velocity, see Figure 1b where

their data is shown. However, in their study the latent heating of growing ice crystals is neglected which gets increasingly

influential on the homogeneous nucleation event for warm temperatures above about $230\,\mathrm{K}$.

In this study, we address these questions by considering the detailed two-moment bulk ice microphysics scheme by Spichtinger and Gierens (2009) as implemented in the numerical model CLaMS-Ice. We vary the parameterization of the saturation vapor pressure and the method for computing the water activity, which is directly linked to the homogeneous nucleation according to Koop et al. (2000). Since the study is based on the numerical model CLaMS-Ice the quantitative aspects of the results do

depend on the model; the qualitative results however are valid in general because each tested parameterization is based on empirical evidence. While our study is focused on the impact of the different formulations for the saturation vapor pressure or the water activity, the study by Spichtinger et al. (2021) investigates the impact of different formulations of the homogeneous nucleation rate coefficient on the simulated homogeneous nucleation by using a slightly different formulation of the two moment ice microphysics. The study is organized as follows: Section 2 describes Koop's approach for homogeneous nucleation

while Section 3 briefly describes the numerical model CLaMS-Ice, which is employed in this study. Section 4 contains the results and the paper ends with some concluding remarks in Section 5.

## 2   Homogeneous Ice Nucleation

In light of the deviating observations of the ice onset humidities (i.e. large supersaturations at ice onset) at the AIDA cloud chamber compared to the prediction of Koop's approach, it is worthwhile to introduce this approach in more detail. Section

2.1 provides an overview and gives a reasonable description of the homogeneous freezing process. Sections 2.2 and 2.3 outline several parametrizations of the saturation vapor pressure of water and the water activity, both of them are needed in Koop's approach. Section 2.4 describes the term "freezing threshold", being often used in literature.

### 2.1   Koop's Approach

The first essential aspect of Koop's approach (see, e.g., Koop et al., 2000; Koop, 2004, 2015) is to express the spontaneous

freezing of the solution particles as a stochastic process, i.e. a solution particle with volume $V$ will freeze within a timespan $t$ with probability

$$P = 1 - \exp(-JVt), \tag{1}$$

where $J$ is the so-called homogeneous nucleation rate coefficient[1]. From the equation it is evident, that a solution particle admits a higher probability of freezing the larger it is. An illustration of the nucleation rate coefficient follows shortly after a

few more details are given.

The second essential step in Koop's approach is to express the nucleation rate coefficient $J = J(\Delta a_w)$ as a function of $\Delta a_w = a_w - a_w^i$, where $a_w$ and $a_w^i$ are the activities of water in the solution and the activity of water in a solution in equilibrium

---

[1]Note, in the atmospheric science literature, $J$ is also referred to as the "nucleation rate", although its unit is per time and volume instead of per time.

with ice, respectively. The former may be written as

$$a_w = \frac{p_{\text{sol}}}{p_{\text{sat}}} \tag{2}$$

with the saturation vapor pressure $p_{\text{sol}}$ of water above the solution and $p_{\text{sat}}$ the saturation vapor pressure of pure liquid water. The saturation vapor pressure $p_{\text{sol}}$ of water above the solution depends on the substance that is dissolved in the particle and its concentration.

A parameterization for the ice activity $a_w^i$ is given in Koop et al. (2000), where this quantity is expressed using a fit to a difference of chemical potentials. Investing a little more theory on thermodynamics (Koop, 2002), the ice activity $a_w^i$ may be 105 rewritten as

$$a_w^i = \frac{p_{i,\text{sat}}}{p_{\text{sat}}}, \tag{3}$$

where $p_{i,\text{sat}}(T)$ is the saturation vapor pressure over an ice surface. The parameterization of $p_{i,\text{sat}}(T)$ that is used in this study and is implemented in CLaMS-Ice is taken from Murphy and Koop (2005), see also Section A1.

Although a precise understanding of the composition of the aqueous solution particles in the upper troposphere is still miss-110 ing, a common assumption for the composition of the solution particles is a binary $H_2O/H_2SO_4$ solution of water and sulfuric acid (Krämer et al., 2006; Minikin et al., 2003). In this case, the saturation vapor pressure $p_{\text{sol}}$ is a function of temperature $T$ and the amount of substance dissolved in the solution particle, e.g. expressed as the mass-fraction $0 \leq x \leq 1$, where $x = 0$ refers to pure water and $x = 1$ to the pure solute (sulfuric acid in this study).

The reason why Koop's approach is widely used in atmospheric science is the following consideration: if a solution particle 115 prevails in the atmosphere, the saturation vapor pressure $p_{\text{sol}}(x,T)$ for the particle should equal the ambient partial pressure of water vapor $p_v$, since otherwise the particle would evaporate or grow to its equilibrium size according to Köhler theory (Köhler, 1922, 1936; Lamb and Verlinde, 2011). In essence, the aqueous solution particles are assumed in equilibrium with their environment. In this case, the water activity (2) reduces to

$$a_w = \frac{p_{\text{sol}}(x,T)}{p_{\text{sat}}(T)} = \frac{p_v}{p_{\text{sat}}(T)} = S_i \frac{p_{i,\text{sat}}(T)}{p_{\text{sat}}(T)} = a_{w,\text{eq}} = \frac{\text{RH}_w}{100\,\%}, \tag{4}$$

where $S_i = \frac{p_v}{p_{i,\text{sat}}(T)}$ is the water vapor saturation ratio (with respect to ice). Note, the Kelvin correction term for curvature is neglected in (4), rendering the expression for $a_w$ independent of the particle size and the solute mass fraction $x$ of the dissolved substance and, thus, also the details of the composition of the solution particles. All these informations are hidden in the saturation ratio $S_i$. In this respect, (4) describes the required saturation ratio such that any solution particle with a given composition $x$ that admits the water activity $a_w$ is in equilibrium with its environment. According to Koop (2015), this 125 assumption is justified for most atmospheric situations "except for very strong updrafts and violent atmospheric wave activity, or when the haze particles are in a highly viscous or even glassy state". Note, there is evidence that the precise composition of the particle is less important as long as it contains anorganic material (Koop et al., 2000; Koop, 2004).

A parameterization of $J$ as a function of $\Delta a_w$ is provided in Koop et al. (2000) as

$$\log_{10} J = P_3(\Delta a_w), \tag{5}$$

where $P_3(\Delta a_w) = b_0 + b_1 \Delta a_w + b_2 (\Delta a_w)^2 + b_3 (\Delta a_w)^3$ is a polynomial of degree 3 in $\Delta a_w$ with real coefficients $b_0, b_1, b_2, b_3$. Although the polynomial $P_3$ is given as a third order polynomial, it nearly coincides with an increasing linear function in the relevant range $0.26 < \Delta a_w < 0.34$, see Figure 2a and Spichtinger et al. (2021). In any case, as $\Delta a_w$ increases, also the values $P_3(\Delta a_w)$ and $J(\Delta a_w) = 10^{P_3(\Delta a_w)}$ increase monotonically.

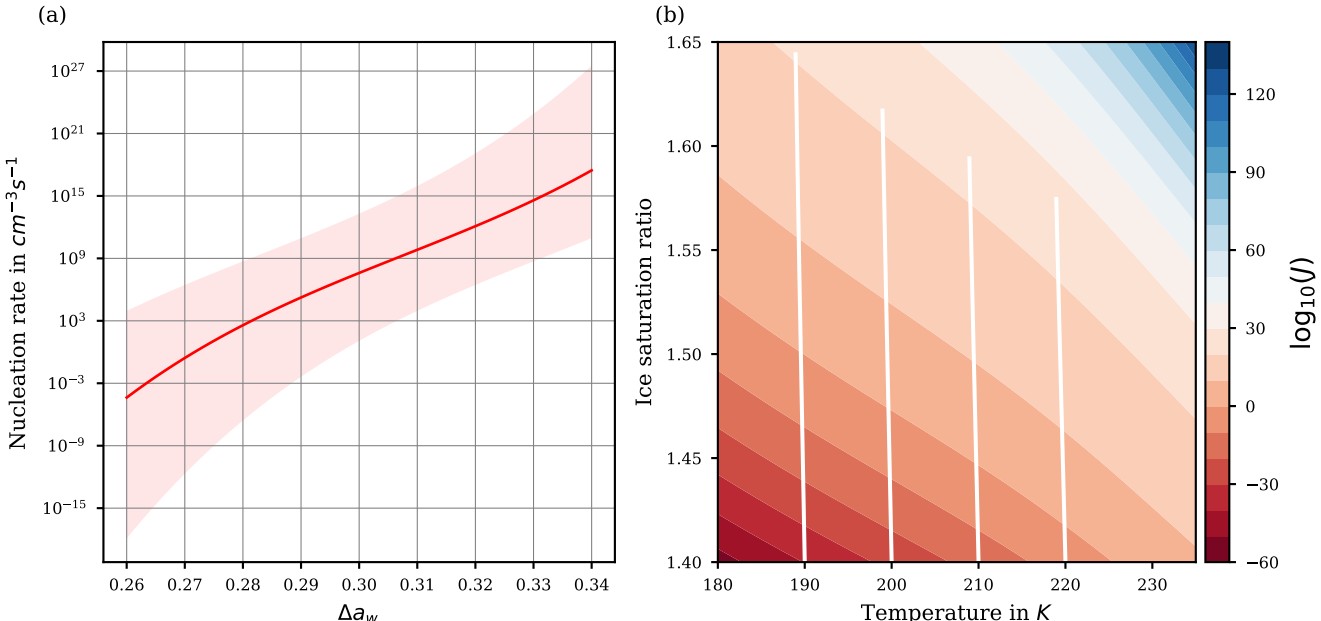

**Figure 2.** Panel a: Homogeneous nucleation rate coefficient $J$ as a function of the difference $\Delta a_w$ of water activities. The light-red shaded area indicates the range of values of $J$ if the $\pm 2.5\%$ uncertainty of the water activity data is taken into account, see Koop (2004); Koop et al. (2000). Panel b: $\log_{10}(J(S_i, T))$ as a function of temperature and environmental humidity $S_i$ by adopting the equilibrium assumption for the solution particles. The white curves show the evolution of temperature and humidity for an air parcel ascending for $1\,\mathrm{h}$ with velocity $0.03\,\mathrm{m\,s^{-1}}$, starting at the bottom.

Adopting the equilibrium assumption for the solution particles (4), together with the expression (3), the difference $\Delta a_w = a_w - a_w^i$ may be rewritten as

$$\Delta a_w = a_w - a_w^i = S_i \frac{p_{i,\mathrm{sat}}(T)}{p_{\mathrm{sat}}(T)} - \frac{p_{i,\mathrm{sat}}(T)}{p_{\mathrm{sat}}(T)} = (S_i - 1) \frac{p_{i,\mathrm{sat}}(T)}{p_{\mathrm{sat}}(T)} \tag{6}$$

and the nucleation rate coefficient $J = J(\Delta a_w)$ becomes exclusively a function of environmental humidity $S_i$ and temperature $T$. Figure 2b shows the nucleation rate coefficient $J(S_i, T)$ as a function of temperature and environmental humidity $S_i$. Obviously, the nucleation rate coefficient increases considerably for increasing humidity. For illustration, the white lines show

values of temperature and humidity for an adiabatically ascending air parcel with vertical velocity $0.03\,\mathrm{m\,s^{-1}}$ and ascending for $1\,\mathrm{h}$, where the initial values are at the bottom of the panel. It is clearly visible, that the saturation ratio and the nucleation

rate coefficient increase. Note, the white lines are slightly tilted to the left due to the adiabatic cooling of the air parcel; the tilt would increase for an increased vertical velocity.

In the following, some additional comments to Koop's approach are listed:

– According to Koop (2004), the water activity data carry an uncertainty of about $\pm 2.5\%$, which may reach values up to $\pm 5\%$, motivating us to include a shaded area around the curve of the homogeneous nucleation rate coefficient in Figure 2a to account for this uncertainty.

    – As already visible in Figure 1a, the solid and dashed freezing threshold lines are not identical

One reason is that the lines computed in Koop et al. (2000) are based on their parameterization of the saturation vapor
pressure $p_{\text{sat}}$ which is not identical to $p_{\text{sat,MK}}$ from Murphy and Koop (2005)[2]. Later, it was recommended to use $p_{\text{sat,MK}}$ (Koop and Zobrist, 2009). The influence of using $p_{\text{sat,MK}}$ on $J$, which produces slightly different combinations of temperature and water activity, is discussed in Koop and Zobrist (2009); Knopf and Rigg (2011); Riechers et al. (2013).

    – The assumption of thermodynamical equilibrium between the gas and the condensed phases is a foundation of Koop's
approach. In general, the updraft velocity of a given air parcel determines the timescale of the sulfuric acid aerosol particles to reach the equilibrium (Knopf et al., 2018; Charnawskas et al., 2017; Berkemeier et al., 2014), but at cold temperatures it is not clear if the particles are indeed in thermodynamical equilibrium. However, the water activity based homogeneous nucleation theory can also cover kinetic effects, i.e. is able to also describe some non-equilibrium cases (Knopf et al., 2018; Murata and Tanaka, 2013; Bullock and Molinero, 2013).

– At the low temperatures at cirrus levels the sulfuric acid particles might have a significant higher visocity that can translate into a higher freezing humidity; an effect that is known for citric acid (Murray et al., 2010; Murray, 2008; Knopf et al., 2018; Wang et al., 2012; Berkemeier et al., 2014; Zobrist et al., 2008). One can even not exclude another phase transition to sulfuric acid monohydrate at these extreme conditions (Koop et al., 2011).

These comments are meant to hint the reader on various problems and issues that arise in the context of the homogeneous
freezing, in particular at cold temperatures. Note, these aspects are usually not represented in numerical models; this is also true for the model CLaMS-Ice that is used in this study and is described further in Section 3.

## 2.2   The Water Vapor Saturation Pressure

The seemingly simple substance "water" admits several anomalies and its properties are still not fully understood. In particular, the properties of supercooled water remain mysterious (Koop, 2004) and experimental evidence is sparse or even unavailable at
temperatures colder than $230\,\text{K}$ (Nachbar et al., 2019). As a result, the saturation vapor pressure over liquid water is well-known at warm temperatures, but bears significant uncertainties at cold temperatures. Consequently, numerous parameterizations for

---

[2]Since we employed $p_{\text{sat,MK}}$ to convert $a_w^i$ to a saturation ratio in Figure 1a, the dashed lines in our plot look more curved compared to the lines in Koop et al. (2000) although our lines are based on the equations 1 and 2 from Koop et al. (2000).

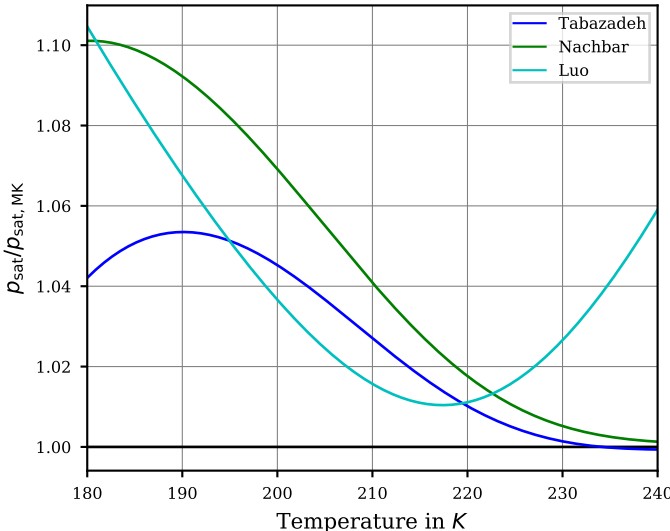

**Figure 3.** Ratio of the saturation vapor pressure parameterizations by Tabazadeh et al. (1997, blue), Nachbar et al. (2019, green), and Luo et al. (1995, cyan) with the widely used parameterization by Murphy and Koop (2005) as functions of temperature. The black line indicates the value 1 for the ratio, i.e. the value of $p_{\mathrm{sat}}$ equals the value of $p_{\mathrm{sat,MK}}$. Note, the formulas of these parameterizations are collected in Appendix A2.

the saturation vapor pressure $p_{\mathrm{sat}}$ exist and their values deviate significantly at cold temperatures as do the experimentally determined datapoints (Murphy and Koop, 2005; Nachbar et al., 2019). A review of classical parameterizations is provided in Murphy and Koop (2005), along with their own parameterization, being valid for temperatures from $123$ to $332\,\mathrm{K}$. Recently,
a new fit for temperatures larger than $200\,\mathrm{K}$ is given in Nachbar et al. (2019). This new parameterization is based on the assumption that ultraviscous water represents a thermodynamical phase of its own, in contrast to the assumption in Murphy and Koop (2005).

In the sequel, the saturation vapor pressure parameterizations

$$p_{\mathrm{sat,MK}}, \quad p_{\mathrm{sat,Tab}}, \quad p_{\mathrm{sat,Nach}}, \quad p_{\mathrm{sat,Luo}} \tag{7}$$

according to Murphy and Koop (2005); Tabazadeh et al. (1997); Nachbar et al. (2019); Luo et al. (1995), respectively, are considered which are widely used in numerous studies and models. Noteworthy, the formulation $p_{\mathrm{sat,Luo}}$ is constructed as a simplification for the expressions from comprehensive thermodynamical models (Pitzer, 1991; Clegg et al., 1994) and is the parameterization that was originally implemented in the CLaMS-Ice model which is employed in this study, see Section 3 below.

In order to assess the differences between these parameterizations, Figure 3 shows the ratios $\frac{p_{\mathrm{sat,Tab}}}{p_{\mathrm{sat,MK}}}, \frac{p_{\mathrm{sat,Nach}}}{p_{\mathrm{sat,MK}}}, \frac{p_{\mathrm{sat,Luo}}}{p_{\mathrm{sat,MK}}}$. Evidently, the values are mostly larger than one, implying that the values of $p_{\mathrm{sat,Tab}}$, $p_{\mathrm{sat,Nach}}$, and $p_{\mathrm{sat,Luo}}$ are larger as the

values of $p_{\mathrm{sat,MK}}$. In summary, from Figure 3 the relation

$$p_{\mathrm{sat,MK}} < p_{\mathrm{sat,Tab}} < p_{\mathrm{sat,Nach}} \tag{8}$$

holds for temperatures from $180$ to $230\,\mathrm{K}$. This relation partly explains the motivation to retain the comparatively old parame-
terization $p_{\mathrm{sat,Tab}}$ since its values are in between the more recent formulations. Moreover, the formulation $p_{\mathrm{sat,Tab}}$ is still used
in other numerical models.

The formulas for the various parameterizations of the saturation vapor pressure over liquid water are summarized in subsection A2.

## 2.3 The Water Activity

### 2.3.1 Formulations

To calculate the water activity $a_w(x,T)$ within a numerical model, there are the following three different possibilities:

(i) Computing the water activity by assuming the solution particles to be in equilibrium with their environment, i.e. using
formula (4) to set $a_w = a_{w,\mathrm{eq}} = \mathrm{RH}_w/100\,\%$, which corresponds to Koop's approach.

(ii) Parameterizing $p_{\mathrm{sol}}$ and computing

$$a_w(x,T) = \frac{p_{\mathrm{sol}}(x,T)}{p_{\mathrm{sol}}(0,T)} \tag{9}$$

by following the definition of the water activity. This formulation relies on a suitable parameterization for $p_{\mathrm{sol}}$. Common
parameterizations of the latter are given in Luo et al. (1995) and Tabazadeh et al. (1997). The parameterization from Luo
et al. (1995) is based on Clegg et al. (1994), while the parameterization from Tabazadeh et al. (1997) is based on the
aerosol thermodynamic model of Clegg and Brimblecombe (1995). For later reference, the notation

$$a_{w,\mathrm{Luo}}(x,T) = \frac{p_{\mathrm{sol,Luo}}(x,T)}{p_{\mathrm{sol,Luo}}(0,T)} \tag{10}$$

for the water activity is introduced, where $p_{\mathrm{sol,Luo}}$ refers to the parameterization from Luo et al. (1995) of the saturation
vapor pressure over a binary $H_2O/H_2SO_4$ solution.

(iii) Parameterizing the function $a_w(x,T)$ directly. Such a formulation is given in Shi et al. (2001), based on the aerosol
thermodynamic model of Carslaw et al. (1995), and henceforth is indicated by the notation

$$a_{w,\mathrm{Car}}(x,T) \tag{11}$$

Noteworthy, also (11) is formulated for a binary $H_2O/H_2SO_4$ solution. The formula of the parameterization (11) is given
in subsection A4.

Here, we introduce the possibilities (i) to (iii) in order to investigate the influence of the $a_w$ formulation on the freezing
threshold and the number of nucleated ice crystals. Note, caution is needed in the actual implementation of the formulations
of $a_w$, since the formulations are based on a specific choice for the saturation vapor pressure in their derivation. Appendix B
outlines the necessary corrections.

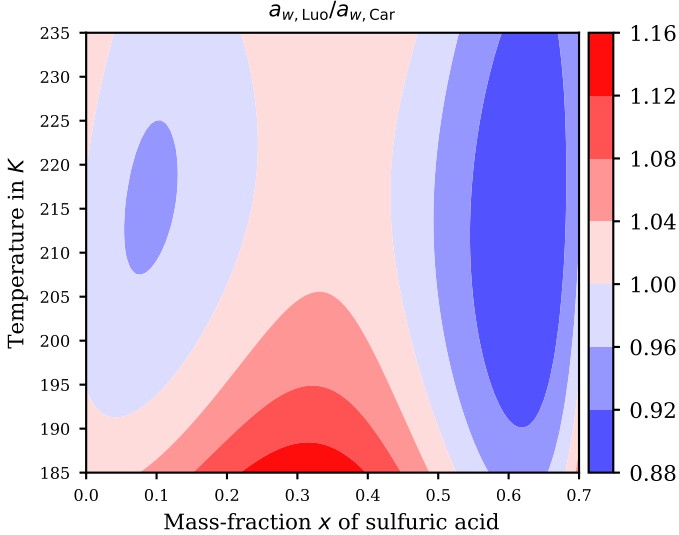

**Figure 4.** Ratio of the formulations (ii) and (iii) (i.e. formulas (B2) and (B3)) for the water activity after Luo and Carslaw as a function of temperature $T$ and mass-fraction $x$ of sulfuric acid.

### 2.3.2 Comparison

After specifying the possibilities (ii) and (iii), i.e. Luo's and Carslaw's formulation for the water activity, the question arises how their values agree. Figure 4 shows the ratio of the actual implemented expressions (B2), (B3) as a function of temperature

$T$ and mass-fraction $x$ of sulfuric acid. Comparing these formulations using their ratio proves beneficial, since then the target formulation $p_{\mathrm{sat}}$ cancels and the comparison is independent of this choice, cf. Appendix B. Note further, the ranges of temperature and mass-fraction in this figure are limited to the specified range of validity of $p_{\mathrm{sol,Luo}}$ given in Luo et al. (1995); the deviation of $p_{\mathrm{sol,Luo}}$ from $a_{w,\mathrm{Car}}(x,T)p_{\mathrm{sat,Car}}(T)$ increases significantly as $x \to 1$.

Evidently, at cold temperatures the ratio is mostly larger than unity (reddish colors), in particular at low to moderate values

of the mass-fraction $x$. At warmer temperatures, the ratio changes from below unity (blue colors) to above unity (red colors) and back for increasing mass-fractions. To interpret the consequences of a ratio below or above unity, consider a region of reddish color, i.e. where $a_{w,\mathrm{Luo}}(x,T) > a_{w,\mathrm{Car}}(x,T)$. Since the homogeneous nucleation rate coefficient $J$ increases monotonically for increasing values of the difference of the water activities (see Figure 2a), the relation $a_{w,\mathrm{Luo}}(x,T) > a_{w,\mathrm{Car}}(x,T)$ implies $J(a_{w,\mathrm{Luo}}) > J(a_{w,\mathrm{Car}})$. Consequently, for a given sulfuric acid solution particle with volume $V$, temperature $T$, and

composition $x$, the relation

$$1 - \exp(-J(a_{w,\mathrm{Luo}})Vt) > 1 - \exp(-J(a_{w,\mathrm{Car}})Vt) \tag{12}$$

for the freezing probabilities results. Thus, the increased freezing probability (12) for Luo's parameterization of the water activity compared to Carslaw's parameterization implies a tendency for earlier freezing of the solution particles, hence we

expect a lower ice onset humidity. Note, earlier freezing of the solution particles in an ascending, thus adiabatically cooling,
air parcel translates to a freezing onset at higher temperatures.

## 2.4 Freezing Thresholds

As explained in Section 2.1, assuming the solution particles in equilibrium with their environment, the water activity $a_{w,\text{eq}}$ is related to a saturation ratio $S_i$ according to Equation (4). This *freezing threshold* is not to be interpreted as a strict limiting value for the humidity at which all solution particles freeze but as an estimate. To clearly show that the freezing threshold is
only an estimate, we subsequently recall its derivation from Koop et al. (2000).

As the first step, for a given volume $V$ of a solution particle, the authors chose the rate $JV = 1\,\text{min}^{-1}$, such that the probability of freezing within $t = 1\,\text{min}$ for these particles is $P = 1 - \exp(-JVt) = 1 - \exp(-1) \approx 0.63$, i.e. about two-thirds of the particles with volume $V$ are expected to be frozen within $1\,\text{min}$. Since $J = J(\Delta a_w)$ and $\Delta a_w = a_w - a_w^i$, one can compute the corresponding critical value $a_{w,\text{eq,crit}}$ of the water activity for any temperature $T$. Using the relation (4), this
critical water activity $a_{w,\text{eq,crit}}$ corresponds to a critical saturation ratio $S_{i,\text{crit}}$. The resulting function $T \mapsto S_{i,\text{crit}}$ for a fixed volume $V$ is commonly referred to as the freezing threshold according to Koop. Note, the variation of $S_{i,\text{crit}}$ in response to a different choice of the radius $r$ of a spherical solution particle, i.e. a volume $V = \frac{4}{3}\pi r^3$, is rather small, see Figure 1a.

Figure 5a shows the critical value $a_{w,\text{eq,crit}}$ of the water activity for spherical solution particles with radii $r \in \{1\,\mu\text{m}, 500\,\text{nm}, 55\,\text{nm}\}$ (solid, dashed, dotted) as a function of temperature. More precisely, the critical water activity is computed as the solution of
the equation $J\left(a_w - a_w^i\right) = \frac{1\,\text{min}}{V}$, where the ice activity $a_w^i$ is either computed by applying the parameterization from Koop et al. (2000) (curves designated as "$a_w^i$ Koop et al (2000)") or its representation as a quotient of saturation pressures according to Equation (3) (curves designated as "$a_w^i$ Quotient"). The small distance between the solid and dotted curves imply that the critical water activity is not very sensitive to the size of the solution particles (a fact that was also remarked in Koop et al., 2000). This motivates to only consider the radius $r = 500\,\text{nm}$ for the solution particles in panels (b) and (c) of this figure.
Observe, the critical water activity increases for decreasing solution particle radius, resembling the required increase in particle volume $V$ to retain a constant value for the product $JV$ contained within the freezing probability, cf. (1). Note, the dependency of the critical water activity on the size of the solution particle implies the fact that a given size distribution of solution particles will be nucleated to ice by homogeneous freezing from the larger to the smaller particles, i.e. the large particles freeze first. Note further, the dependency of $a_{w,\text{eq,crit}}$ on temperature is stronger.
According to Equation (4), any value of the water activity can be translated into a saturation ratio. Figure 5b shows the critical saturation ratio $S_{i,\text{crit}}$ corresponding to the critical water activity $a_{w,\text{eq,crit}}$ from panel (a) for radius $r = 500\,\text{nm}$ as computed with the various parameterizations for the saturation vapor pressure of water described in Section 2.2. The magenta curves are based on $p_{\text{sat,MK}}$ and thus are the same as in Figure 1a (note the meaning of the solid and dashed lines in the two figures is interchanged). The other freezing curves show systematically larger values at cold temperatures in comparison to the magenta
curves. The reason for these deviations are the different values of the various saturation vapor pressures parameterization, i.e. a larger value for the saturation vapor pressure results in a larger freezing threshold, cf. Figure 3.

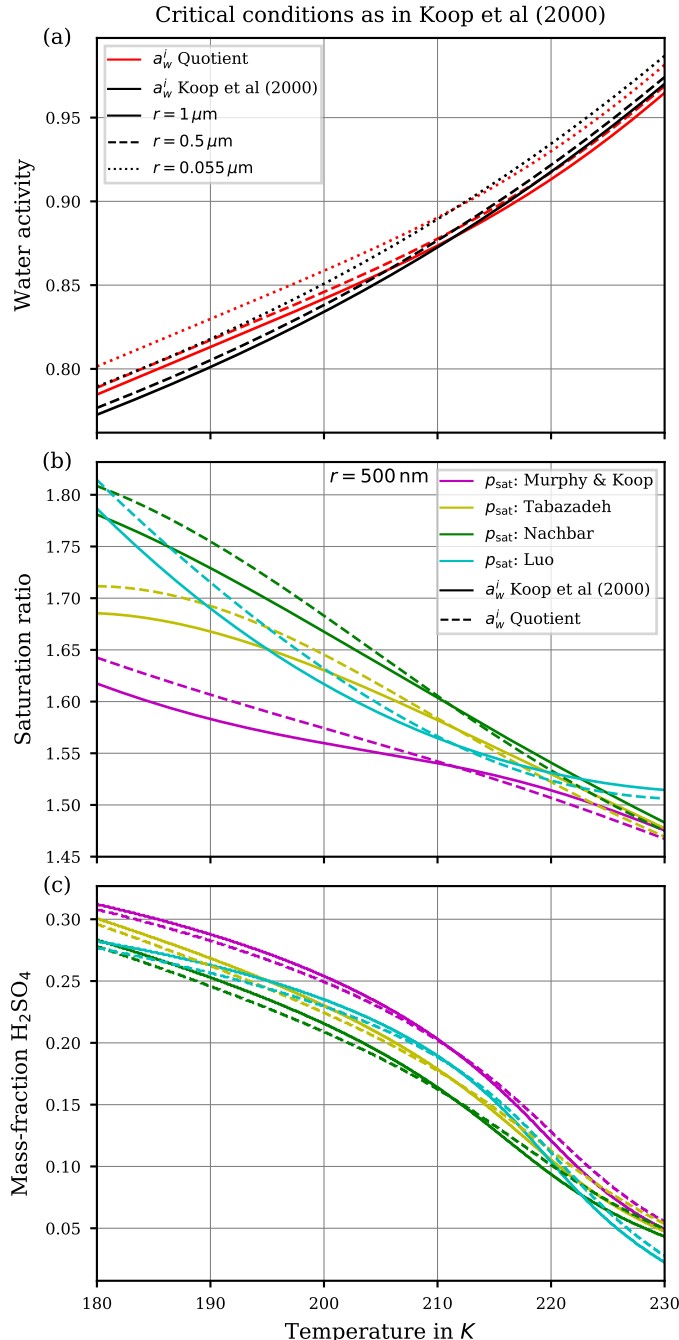

**Figure 5.** Critical conditions for the freezing threshold computations motivated by Koop et al. (2000). The curves designated as "$a_w^i$ Koop et al (2000)" refer to the computations with $a_w^i$ taken from Koop et al. (2000) wheres the curves designated as "$a_w^i$ Quotient" indicate the computation of $a_w^i$ as the quotient of saturation pressures (3). Panel (a) shows the critical value of the water activity for various sizes of the spherical solution particles. Panel (b) comprises the critical saturation ratio, i.e. converting the critical water activities from panel (a) into a saturation ratio as in (4). Panel (c) shows the required mass-fraction to realize the critical value of the water activity shown in panel (a).

Figure 5c comprises, as a function of temperature and with the same markers and colors as in panel (b), the sulfuric acid mass-fraction $x_{\mathrm{crit}}$ that is needed, such that the corresponding value of the water activity coincides with the critical water activity $a_{w,\mathrm{eq,crit}}$ shown in panel (a). Note, the lower the value of the critical mass-fraction $x_{\mathrm{crit}}$, the more diluted the solution particle, i.e. a value zero indicates a pure water particle. As an example, consider the temperature $T = 190\,\mathrm{K}$. Focusing on the formulation of $a_w^i$ according to Koop et al. (2000), the critical value for the water activity is about $0.8$ (solid black curve in panel a). The required mass-fraction to achieve this value varies between $0.25$ and $0.29$ (range of solid curves in panel c at $T = 190\,\mathrm{K}$), depending on the parameterization of the saturation vapor pressure. In particular, using $p_{\mathrm{sat,MK}}$ (magenta curve) needs a larger mass-fraction in comparison to $p_{\mathrm{sat,Nach}}$ (green curve). In other words, using $p_{\mathrm{sat,MK}}$ requires less diluted solution particles than for $p_{\mathrm{sat,Nach}}$.

As is often done in detailed numerical models, cf. Section 3, a *dry aerosol distribution* is prescribed and, for a given temperature, the model computes the size of the solution particles at the current temperature and humidity as predicted by Köhler theory[3]. Thus, internally the dry aerosol particles are converted into solution particles. In the next step their freezing probability is determined. For the example outlined above, the predicted freezing is delayed by using $p_{\mathrm{sat,Nach}}$, i.e. at higher saturation ratios $S_i$, since the solution particles need to be more diluted as compared to $p_{\mathrm{sat,MK}}$.

However, we emphasize that all of the curves displayed in Figure 5 depend crucially on the choice of the value for the rate $JV$ which in this case is $JV = 1\,\mathrm{min}^{-1}$ to resemble the choice from Koop et al. (2000). Choosing a different rate obviously results in different curves. In particular, the so-called *freezing threshold* does *not* imply, that a given particle will freeze if its environmental humidity equals the critical saturation ratio; it only implies that the particle now has a probability of about $63\,\%$ to freeze within the next minute. More precisely, numerical and analytical analyses of homogeneous freezing experiments in the box-model context implementing Koop's approach for nucleation show that the actual freezing of the solution particles already starts at lower values of the saturation ratio (Baumgartner and Spichtinger, 2019; Spichtinger et al., 2021). However, the aforementioned choice $JV = 1\,\mathrm{min}^{-1}$ leads to freezing thresholds that compare well with observed freezing humidities, at least for warm temperatures (Schneider et al., 2021).

## 3   Description of the Sensitivity Studies

This section contains a short description of the numerical ice microphysics model "CLaMS-Ice" which is used in this study to perform systematic studies analyzing the influence of different parameterizations of the water vapor saturation pressure $p_{\mathrm{sat}}$ and formulations of the water activities $a_w$ on the ice onset humidity and ice crystal concentration during ice nucleation events for various temperatures and updraft velocities.

### 3.1   CLaMS-Ice

CLaMS-Ice is a trajectory model for ice microphysics, intended to compute the ice microphysics along an air parcel trajectory originating from the CLaMS model (*Chemical Lagrangian Model of the Stratosphere*, Pommrich et al., 2014; Konopka et al.,

---

[3]Note, Köhler theory allows to determine the size of the solution particles by assuming them in thermodynamical equilibrium.

2007; Luebke et al., 2016), modified to also accept trajectories as computed from ERA-Reanalysis data (Dee et al., 2011), from AIDA cloud chamber measurements, or artificial trajectories. The CLaMS-Ice model implements the two-moment microphysics as described in Spichtinger and Gierens (2009), which includes homogeneous as well as heterogeneous nucleation of ice, depositional growth of ice crystals, their evaporation, aggregation, and sedimentation. Similarly to other two-moment schemes, the model predicts the ice number concentration and the ice mass mixing-ratio. Internally, the ice population is assumed to follow a lognormal size distribution. In general, the input trajectory needs to provide temporal evolutions of pressure and temperature, together with the initial water content. Since the model solely implements ice microphysics, it is assumed that the initial water is exclusively partitioned between the gas and the ice phase, while liquid water is not treated. Hence the model is only applicable at temperatures below $235\,\mathrm{K}$.

For this study, the various parameterizations of the saturation vapor pressure $p_{\mathrm{sat}}$ and the formulations of the water activity $a_w$ outlined in the preceding Sections 2.2 and 2.3 are implemented in the CLaMS-Ice model.

## 3.2 Homogeneous ice nucleation

In order to allow homogeneus ice nucleation, a dry aerosol population consisting of sulfuric acid particles is prescribed. In this context "dry" means that the particles exclusively consist of sulfuric acid. The size distribution of the dry aerosol particles is assumed to follow a lognormal size distribution

$$f_a(r) = \frac{N_a}{\sqrt{2\pi}\log(\sigma_a)r} \exp\left(-\frac{1}{2}\left(\frac{\log(\frac{r}{r_a})}{\log(\sigma_a)}\right)^2\right) \tag{13}$$

with geometric mean radius $r_a$, geometric standard deviation $\sigma_a$, and the number concentration $N_a$ of dry aerosol particles. In each timestep, the same dry aerosol size distribution is assumed, i.e. only $N_a$ is updated and the values for $r_a, \sigma_a$ are retained, and the model computes the resulting size distribution of solution particles by applying Köhler theory to adjust the size of the solution particles to the ambient humidity. Assuming the same, unchanged size distribution for the dry aerosol in each timestep represents a major modeling assumption of a two-moment microphysics scheme but relieves the model to additionally carry further variables to describe the background aerosol. A consequence of this assumption is discussed in Section 4.1.1. After determining the size of the solution particles, homogeneous ice nucleation is predicted by computing the freezing probability for the solution particles and let them freeze if the freezing probability is large enough, thereby adapting the number of nucleated ice particles. More precisely, given the current model timestep $\Delta t$, the minimal volume $V_{\mathrm{min}}$ of the solution particles is determined for which $JV_{\mathrm{min}}\Delta t \geq 10^{-7}$ and all larger solution particles are assumed to freeze within the current timestep. After their nucleation, the ice crystals are allowed to grow and evaporate by diffusion, and to sediment out of the air parcel, although the latter is not relevant for the nucleation event itself. Note, heterogeneous freezing and aggregation of ice crystals are turned off for the current simulations, since these processes are of minor relevance to the number of nucleated ice crystals during a nucleation event and are not affected by either the saturation vapor pressure or the water activity.

At this stage we hint the reader on an important aspect that needs to be kept in mind when interpreting the freezing of aqueous aerosol particles with the theory outlined in Sect. 2. If the particle size distribution is assumed as lognormal, a value $\sigma_a$ close to

unity implies that most of the aerosol particles have sizes comparable to the geometric mean radius $r_a$. In contrast, for broader size distributions (i.e. larger $\sigma_a$), the geometric mean radius gets less meaningful for the dominant sizes of the aerosol particles. In effect, the sizes of the freezing aerosol particles are (much) larger in comparison to $r_a$. Appendix C illustrates this fact.

### 3.3   Where is the water activity required in CLaMS-Ice?

It is worth to note the two routines within the CLaMS-Ice model where the water activity is utilized:

– the freezing probability of the solution particles is computed within the routine of the homogeneous freezing,

      – the size of the solution particles is computed within the routine applying Köhler theory.

The first location is already discussed in Section 2.3. However, the second occurrence calls for an explanation, i.e. within the routine applying Köhler theory to the growth of the solution particles. According to Köhler theory, the saturation vapor pressure over a spherical solution particle depends on its curvature (Kelvin effect) and its chemical composition (Raoult's

law), see, e.g., Lamb and Verlinde (2011, Section 3.5). Köhler theory describes the equilibrium conditions for a given solution particle, i.e. given the amount of sulfuric acid it predicts the size of the resulting solution particle, hence the amount of water which condenses onto the surface of the particle in order to reach thermodynamical equilibrium given the current environmental conditions (pressure, temperature, and humidity).

    As a consequence of the two occurrences of the water activity in the model, there is some freedom in combining the various

possibilities for its computation. In the following, we fix the use of Luo's method (10) for the computation of the water activity within the Köhler routine. The reason for this choice is discussed together with the consequences of using Carslaw's method (11) within this routine in Section 4.4.

    The freedom in combining the formulations of the water activity at the aforementioned two code locations further allows to also test inconsistent combinations, i.e. using different formulations. It turns out that such an inconsistent combination gives the

best match in comparison to the AIDA observations in terms of the ice onset humidity, whereas the consistent combinations give similar but slightly poorer results (see Figure 14, Section 4.5.1). A possible explanation is by recalling that Koop's approach rests on taking thermodynamic constraints into account together with the assumption $a_w = S_i a_w^i$, i.e. neglecting the Köhler correction, such that the aerosol water activity (and consequently also the nucleation rate coefficient) is set solely by the environmental conditions. Usage of inconsistent formulations may break the link between the environmental conditions

and the nucleation rate from Koop et al. (2000). Mathematically, the water activity $a_{w,2}$ within the freezing routine is no longer set by the environmental conditions but by a non-trivial relation $a_{w,2} = f_{1 \mapsto 2}(a_{w,1}(x,T), T)$ where $a_{w,1}$ is the water activity computed in the Köhler routine. In a consistent setting, the conversion function $f_{1 \mapsto 2}$ equals the identity, whereas in the case of inconsistent formulas the conversion function modifies the water activity $a_{w,1}$ to become $a_{w,2}$ which is then used to determine the freezing. In effect, this could be the reason for larger deviations of the freezing humidities from the Koop-line

in the inconsistent settings in comparison to the consistent ones. Such an non-trivial link function may be interpreted as a "compensating bias" or a "option for tuning" of the model at hand.

| $r_a$ | 25 nm | 55 nm | 100 nm | 500 nm |
|-------|-------|-------|--------|--------|
| $\sigma_a$ | 1.1 | 1.3 | 1.6 | 1.9 |

**Table 1.** Choices of the dry aerosol parameters $r_a$ and $\sigma_a$.

Note, in contrast to the water activity, the parameterization of the saturation vapor pressure is used throughout the model code, e.g. to routinely convert the vapor mixing-ratio $q_v$ into a saturation ratio $S_i$. Consequently, we do not provide a list of all code locations where $p_{sat}$ is utilized and only remark that a change of the parameterization of $p_{sat}$ is applied everywhere in the model code.

### 3.4 Sensitivity Studies

In order to assess the sensitivity of homogeneous ice nucleation on the parameterizations of the saturation vapor pressure of water and the formulation of the water activity, simulations are performed similar to the approach in Kärcher and Lohmann (2002). This approach was already employed in Krämer et al. (2016) but with the numerical model MAID (Bunz et al., 2008), implementing a much more detailed description of the ice microphysics compared to the two-moment scheme from CLaMS-Ice. The approach in Kärcher and Lohmann (2002) consists in the use of artificial adiabatic air parcel trajectories to investigate single homogeneous ice nucleation events. More precisely, air parcel trajectories are created describing an adiabatic ascend with constant vertical velocity, ranging from $0.01\,\mathrm{m\,s^{-1}}$ up to $8\,\mathrm{m\,s^{-1}}$. The initial pressure is always set to $220\,\mathrm{hPa}$ and the initial temperatures are mainly

$$196\,\mathrm{K}, \quad 216\,\mathrm{K}, \quad \text{and} \quad 235\,\mathrm{K}. \tag{14}$$

Note, more temperatures were chosen to illustrate the ice onset humidities as a function of temperature, i.e. initial temperatures in intervals of $4\,\mathrm{K}$ starting at $196\,\mathrm{K}$. Simulations were conducted for various choices of the geometric mean radius $r_a$ and the geometric width $\sigma_a$ of the lognormal size distribution (13) of the dry aerosol particles; these choices are summarized in Table 1.

In terms of number of nucleated ice crystals, all our simulation results are compared to the data described in Kärcher and Lohmann (2002) which serve as a reference. The much more detailed model used in that study is described in Kärcher (2003). There, the parameterization $p_{sat,Luo}$ is employed for the saturation ratio of water, together with the formulation $a_{w,Luo}$, see Equation (10), for the water activity within the homogeneous freezing process. Moreover, the dry aerosol size distribution was also assumed as lognormal with the parameters $r_a = 55\,\mathrm{nm}$ and $\sigma_a = 1.6$ (Kärcher and Lohmann, 2002).

To avoid the nucleation events to exhaust the aerosol population, the number density $N_a$ of the dry aerosol particles is chosen as $10^4\,\mathrm{cm^{-3}}$, in-line with Kärcher and Lohmann (2002). Arguably, this value is unrealistically high and a more realistic value is $300\,\mathrm{cm^{-3}}$ (see, e.g., the choice in Spichtinger and Gierens, 2009), but the focus of the current study is on the influence of the various choices for the parameterization.

A summary of the various simulation scenarios is provided in Table 2 for the reader's convenience.

| Geometric mean radius $r_a$ | Geometric width $\sigma_a$ | Figure reference | Feature/Comment |
|---|---|---|---|
| 55 nm | 1.1 | $N_{\text{ice}}$: Fig. 6 <br> $S_{\text{i,onset}}$: Fig. 9 | Narrow dry size distribution |
| 55 nm | 1.6 | $N_{\text{ice}}$: Fig. 7 <br> $S_{\text{i,onset}}$: Fig. 10 | Broad dry size distribution |
| 55 nm | 1.6 | $S_{\text{i,onset}}$: Fig. 11 | Use $a_w^i = \frac{p_{i,\text{sat}}}{p_{\text{sat}}}$ instead of parameterization <br> from Koop et al. (2000) |
| 25 nm, 55 nm, 100 nm, 500 nm | 1.1 | $N_{\text{ice}}$: Fig. 8 | Effect of increasing $r_a$; <br> only $p_{\text{sat,Nach}}$ |
| 55 nm | 1.1 | $S_{\text{i,onset}}$: Fig. 12 | Use $a_{w,\text{Car}}$ in Köhler equilibrium computation <br> with both methods for $a_w^i$ |

**Table 2.** Summary of the various simulation setups discussed in this study; $N_{\text{ice}}$ refers to a plot of the number of nucleated ice crystals versus the updraft velocity and $S_{\text{i,onset}}$ to a plot of the freezing ice saturation ratio versus temperature.

## 4 Results and Discussion

### 4.1 A General View on the CLaMS-Ice Results

Before discussing more detailed influences resulting from the exchange of the parameterization of the saturation vapor pressure $p_{\text{sat}}$ (Section 4.2) or the method to compute the water activity $a_w$ (Section 4.3), we first expand on more general aspects of the results in Sections 4.1.1 and 4.1.2.

### 4.1.1 Number of nucleated ice crystals

Figure 6 shows the number of homogeneously nucleated ice crystals as a function of the vertical velocity in the same format as in Kärcher and Lohmann (2002) for the choices $r_a = 55\,\text{nm}$ and $\sigma_a = 1.1$, describing a relatively narrow size distribution of small dry aerosol particles. In contrast, Figure 7 shows the same for the choices $r_a = 55\,\text{nm}$ and $\sigma_a = 1.6$, i.e. a broader dry aerosol size distribution with the same geometric mean radius. The different panels in these figures correspond to different choices of the parameterization of the saturation vapor pressure of water used within the individual model runs. Different colors indicate the different initial temperatures given in (14), while the data points indicated by stars represent the data of Kärcher and Lohmann (2002). The other markers, which are connected by straight line segments, comprise the number of nucleated ice crystals by employing the different methods for computing the water activity, i.e. circles correspond to the equilibrium assumption (2), the boxes to Carslaw's method (11), and the crosses to Luo's method (10).

As already discussed in Kärcher and Lohmann (2002); Spichtinger and Gierens (2009); Krämer et al. (2016), the higher the updraft velocity, the higher is the number of homogeneously nucleated ice crystals, since the stronger the updraft the stronger is the adiabatic cooling and the larger is the supersaturation that can be reached within the air parcel. The simulation results

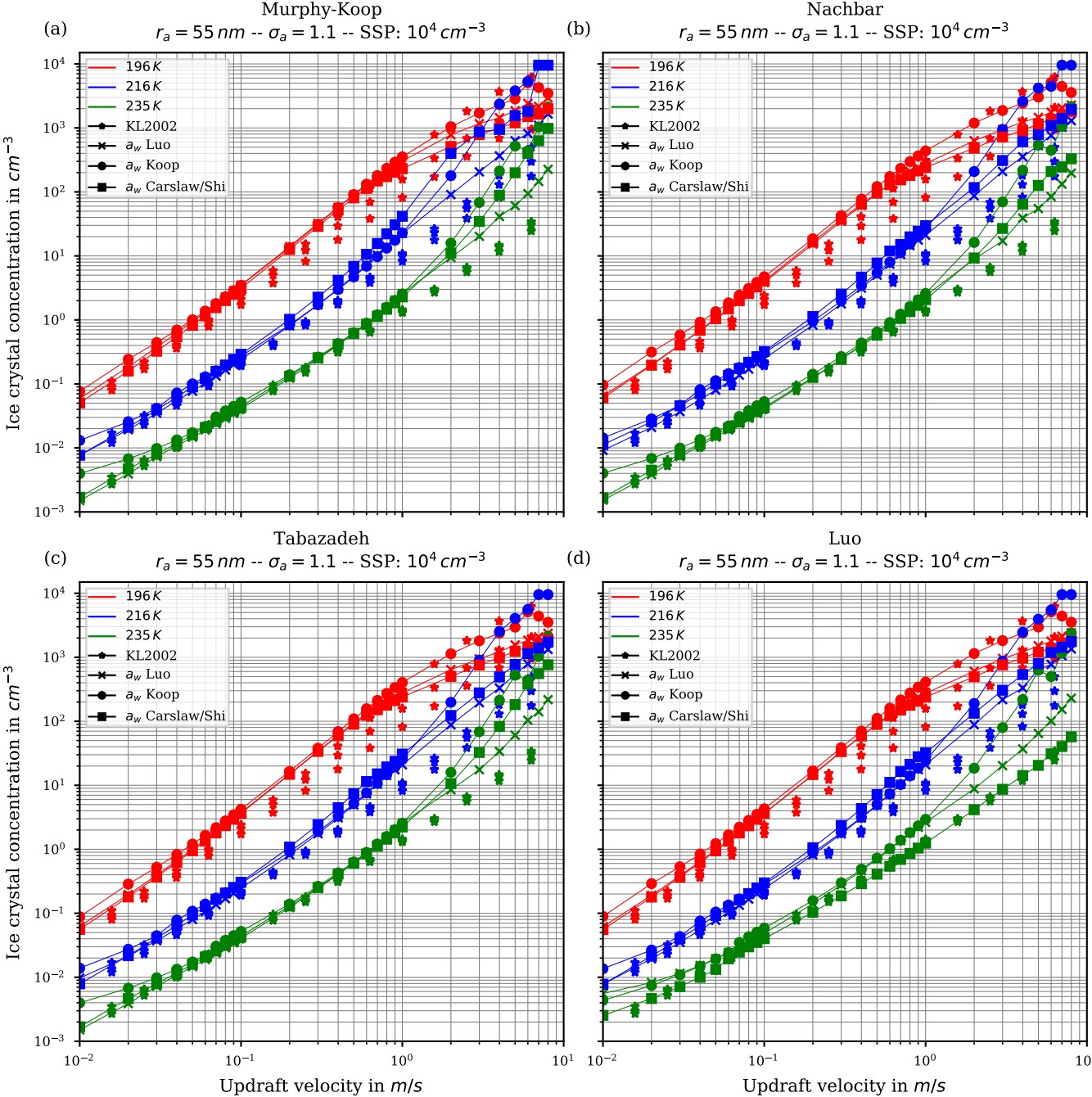

**Figure 6.** Number of homogeneously nucleated ice crystals as a function of the vertical updraft velocity with the choices $r_a = 55\,nm$ and $\sigma_a = 1.1$ for the dry aerosol lognormal size distribution. Different panels correspond to different parameterizations of the saturation vapor pressure (also indicated in the title). Connected symbols indicate the number of nucleated ice crystals for the different formulations for water activity, star-symbols show the data by Kärcher and Lohmann (2002) as in Figure 1b. For further details, see the text. Note, SSP refers to the sulfuric acid aerosol particles (small solution particles).

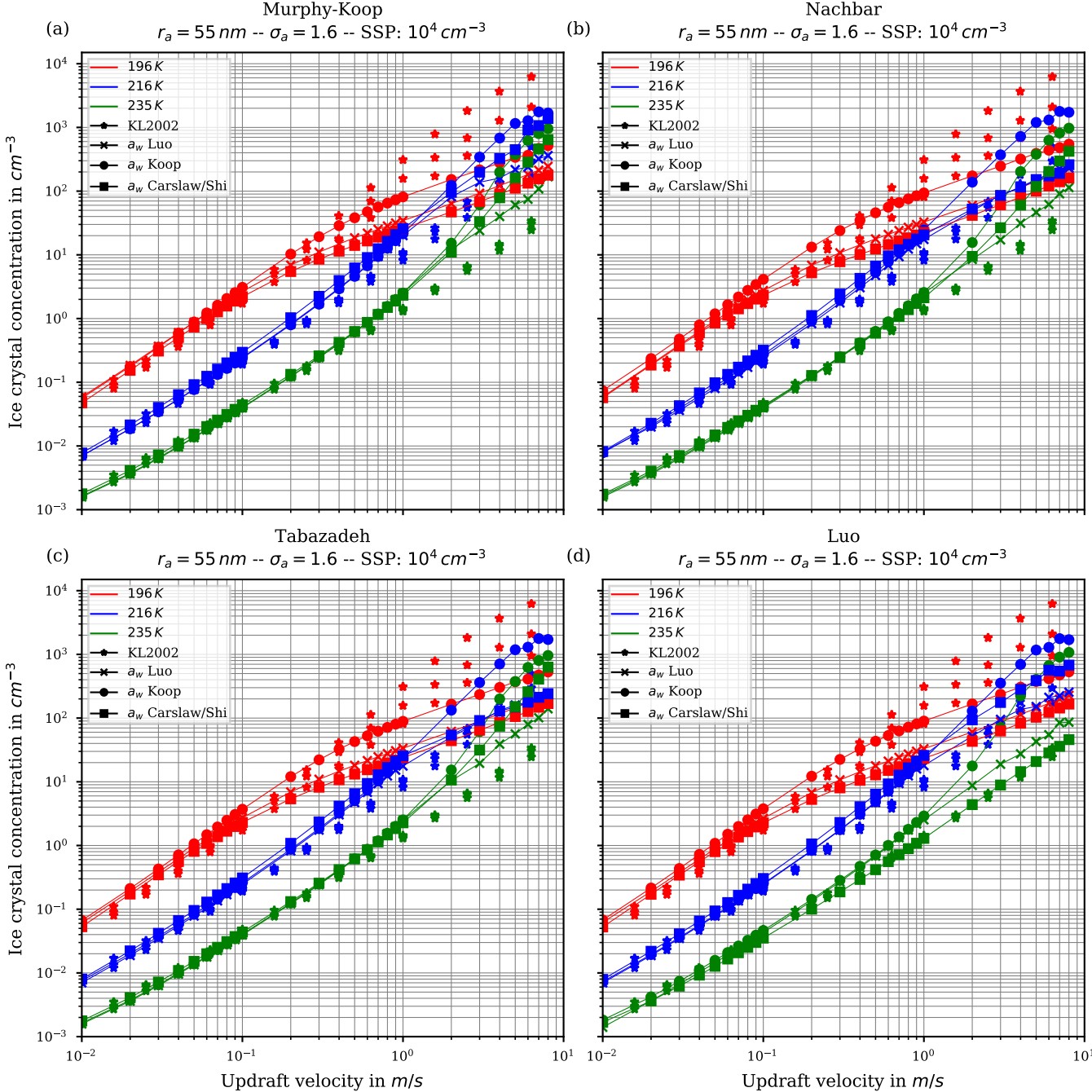

**Figure 7.** The same as in Figure 6 but with the choice $\sigma_a = 1.6$ for the dry aerosol lognormal size distribution.

clearly represent the trend of the data by Kärcher and Lohmann (2002), i.e. the number of nucleated ice crystals increase for increasing vertical velocity. A more detailed discussion of the differences and similarities of the computed results will be given

below.

Another general feature becomes evident by comparing the respective panels in Figures 6 and 7, where only the assumed geometrical width $\sigma_a$ of the dry aerosol distribution is increased. A broader size distribution indicated by a larger $\sigma_a$ results in a decreased number of nucleated ice crystals, most pronounced at large vertical velocities. To explain this dependency on the geometrical width parameter, recall that a larger geometrical width corresponds to a broader size distribution of the dry aerosol.

Consequently, larger solution particles are present, which have a higher probability to freeze in comparison to the smaller ones (see (1)). As a result, the large ice crystals nucleate earlier within the ascending air parcel whereby their diffusional growth diminishes the increase of the supersaturation caused by the adiabatic cooling. Consequently, the maximal supersaturation decreases and in total fewer ice crystals nucleate. These observations are consistent with the results presented in Liu and Shi (2018).

Comparing the number of nucleated ice crystals in the Figures 6 and 7 with the data points from Kärcher and Lohmann (2002, star markers) the general agreement and the general trends are captured well. At low to medium vertical velocities, the CLaMS-Ice model predicts slightly more nucleated ice crystals compared to the reference results at warm temperatures, cf. Figure 6. Since the agreement is better in Figure 7 where a broader aerosol size distribution with $\sigma_a = 1.6$ was used, the slight overestimation stems from the nucleation of smaller ice crystals. However, the much more noticeable feature is the

underestimation of the number of ice crystals at large vertical velocities, in particular at cold temperatures. As explained in Spichtinger and Gierens (2009), this behavior is most likely attributable to a modelling artifact caused by an assumption within the development of the bulk-microphysics scheme as included in CLaMS-Ice in combination with a (too) large timestep. The basic assumption of the bulk-microphysics scheme is that the dry aerosol distribution is the same in every timestep. As soon as homogeneous nucleation starts, this assumption is clearly violated and the number of nucleated ice crystals from the previous

timestep should be subtracted from the total aerosol background. However, this is not included, hence the number of nucleated ice crystals from the previous timestep is forgotten in the subsequent timestep and the number of nucleated ice crystals is overestimated. In addition, at high vertical velocities, the model timestep needs to be decreased significantly to resolve the fast changes in the saturation ratio during the nucleation process. For the current simulations, the CLaMS-Ice model was configured to adaptively use a timestep between $0.1\,\mathrm{s}$ and $1\,\mathrm{s}$, where the timestep is decreased if nucleation takes place. In contrast, the

timestep for the model used in Kärcher and Lohmann (2002) was allowed to decrease to $5\,\mathrm{ms}$. Using the timestep $5\,\mathrm{ms}$ in the CLaMS-Ice simulations alleviates the model artifact at large vertical velocities, see Appendix D.

Before proceeding, we illustrate the influence of increasing the geometrical radius $r_a$ on the number of nucleated ice crystals in Figure 8. In all panels of this figure Nachbar's parameterisation $p_{\mathrm{sat,Nach}}$ is used. The respective figures for the other choices for the parameterization of the saturation vapor pressure are similar and therefore not shown. As is easily seen by comparing

the panels in this figure, increasing the size of the dry aerosol particles results in a decrease in the number of nucleated ice crystals. The explanation is the same as before for the effect of the increase of the geometrical width of the distribution: a larger geometrical radius implies the presence of larger solution particles which freeze earlier in the present scenario of an

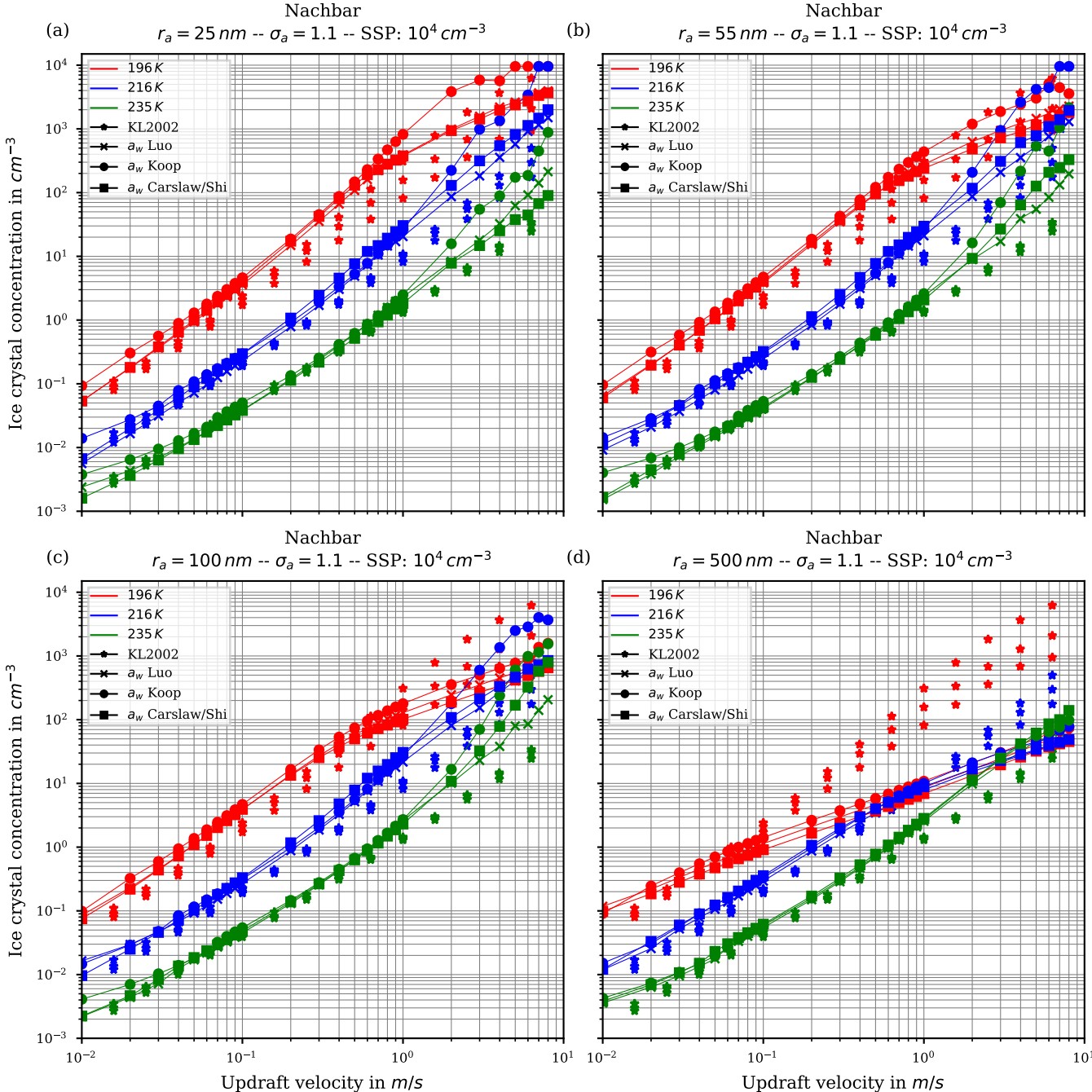

**Figure 8.** Fixing Nachbar's parameterization $p_{\mathrm{sat,Nach}}$ for the saturation vapor pressure. The panels show the influence of increasing the geometrical radius $r_a$ of the lognormal size distribution for the dry aerosol (indicated in the title of each panel) on the number of nucleated ice crystals by keeping the geometrical width $\sigma_a = 1.1$ constant; the format of the panels is as in Figure 6.

ascending air parcel (i.e. at higher temperatures), hence their diffusional growth quenches the nucleation event by diminishing the maximum supersaturation, thus shorten the time of the nucleation event which results in fewer freezing ice crystals.

### 4.1.2 Ice onset humidity

Apart from the number of nucleated ice crystals considered in the previous section, the actual freezing threshold is of interest, i.e. the ice saturation ratio (and temperature) as of when the homogeneous nucleation starts. For the same scenarios as in the Figures 6 and 7, the Figures 9 and 10 show the freezing threshold from the model simulations, commonly referred to as the ice onset. Here, the freezing threshold is defined as the time instant where the model predicts the existence of ice. Note, additional initial temperatures are included in these figures to more clearly show the simulated ice onset humidities as a function of temperature. As before, the individual panels in Figures 9 and 10 correspond to the different choices of the parameterization of the saturation vapor pressure. In contrast, the figures now show the saturation ratio at ice onset (dots) as a function of temperature The colors indicate the employed method to evaluate the water activity: red according to Luo's method, blue by assuming equilibrium (Koop), and green according to Carslaw's method. Note, within these plots, there are dots for each scenario, i.e. for each initial temperature and updraft velocity. The different temperature choices are clearly visible, but the ice onset humidities for the different vertical velocities are very close to each other and merely collapse to a single dot as expected, because the ice onset is only influenced by the volume of the solution particles (see Section 2.1) which is independent of the velocity of the air parcel.

The dashed blue line is the "Koop line", i.e. the critical saturation ratio as in Koop et al. (2000) corresponding to the dashed magenta curve in Figure 5b. The light-blue shaded area around this curve indicates the uncertainty that stems from the uncertainty of $\pm 2.5\%$ in the water activity data mentioned in Koop (2004); Koop et al. (2000), see also Figure 2a. Inclusion of this curve is intended as a rough predictor when solution particles will freeze by assuming them in equilibrium and to guide the eye.

The magenta curves are fitted through observational results obtained at the AIDA cloud chamber for sulfuric acid solution particles. The derivation of these curves together with the underlying measurement data is described in Schneider et al. (2021). The dashed magenta curve represents a fit through the measurement data while the solid magenta line additionally takes the constraint into account, that the curve should pass the point $(235\,\mathrm{K}, p_{\mathrm{sat,MK}}(235\,\mathrm{K}))$. The coloured range along these curves span 0.1 in each direction and is intended as an indication of (measurement) uncertainty. A thorough discussion of the uncertainties inherent in the observational data is provided in Schneider et al. (2021).

The solid black line indicates water saturation as computed with the respective choice for the parameterization of the water saturation pressure. The black dotted line represents water saturation as computed with Nachbar's parameterization $p_{\mathrm{sat,Nach}}$ and serves as a reference, since this is the latest parameterization from literature.

Comparing the respective panels in Figures 9 and 10, it becomes visible that increasing the width of the dry aerosol size distribution (as done from Figure 9 to 10) results in a decrease of the ice onset independently of the choice of parameterization of $p_{sat}$. This, again, resembles the influence of the width of the dry aerosol size distribution, where larger solution particles

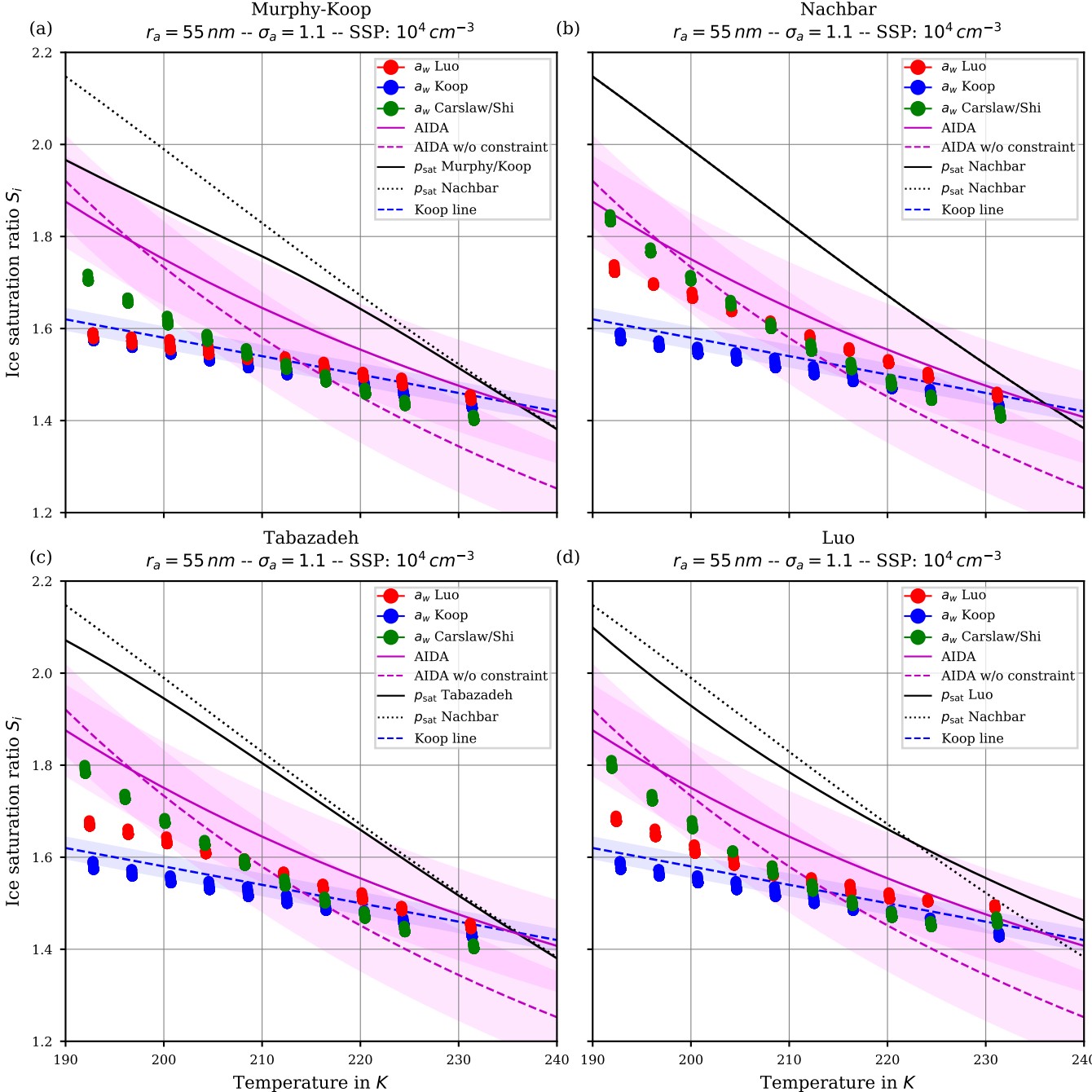

**Figure 9.** Ice onset as a function of temperature for the same scenarios as in Figure 6 with the choice $\sigma_a = 1.1$ for the dry aerosol lognormal size distribution. In this plot, each air parcel simulation with a different updraft velocity is represented as a dot and the colors indicate the employed water activity formulation. Magenta lines indicate fits to the AIDA measurements while the shaded areas show a $\pm 0.1$ uncertainty range. The blue dashed line indicates a Koop line and the black dotted line the values of $p_{\text{sat,Nach}}$. The solid black line shows the values of the saturation vapor pressure formulation indicated in the panel's title. Note, SSP refers to the sulfuric acid aerosol particles (small solution particles).

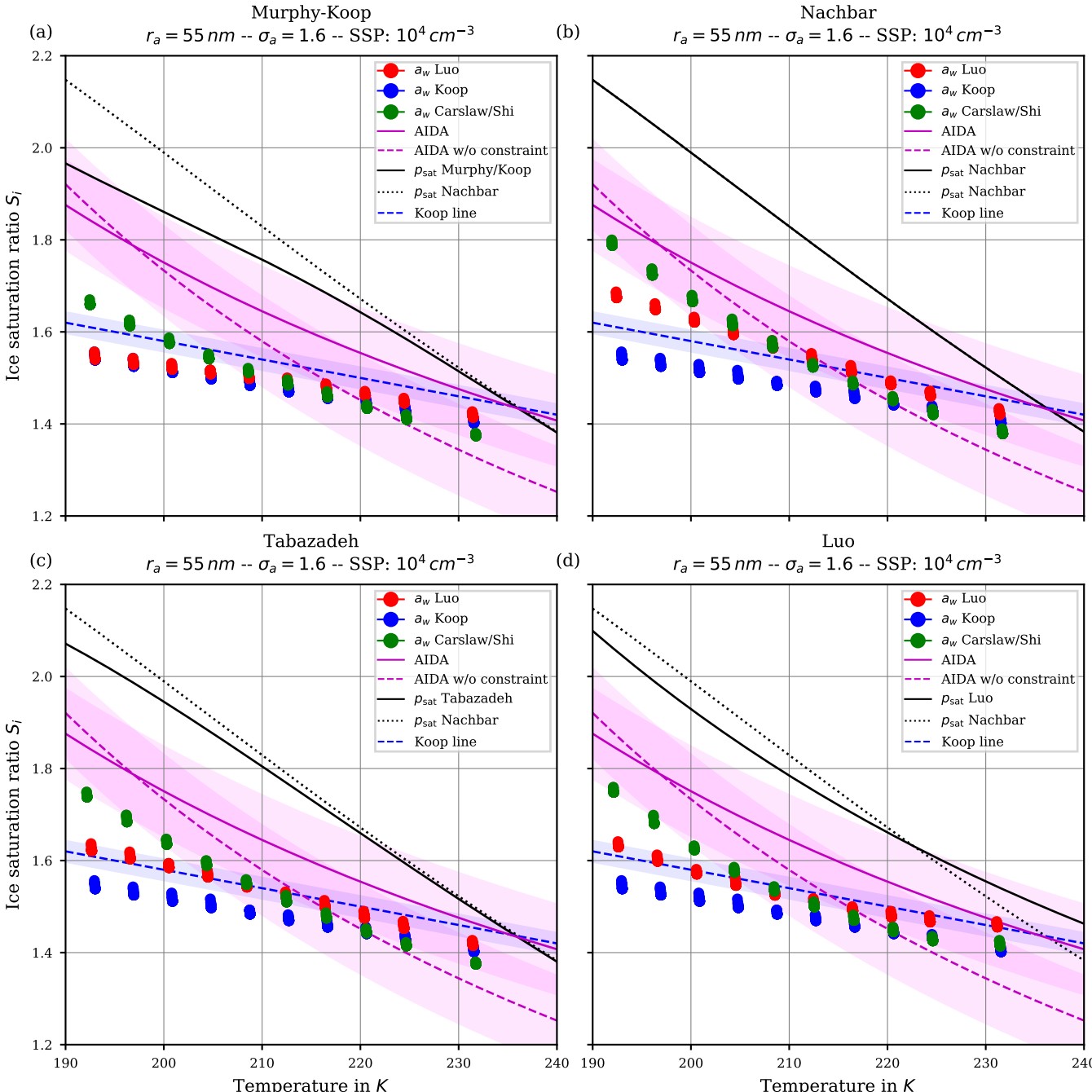

**Figure 10.** The same as in Figure 9 but with the choice $\sigma_a = 1.6$ for the dry aerosol lognormal size distribution.

are present for larger values of $\sigma_a$ which freeze earlier. Since the air parcel trajectories describe an ascending parcel, earlier freezing (i.e. at warmer temperatures) necessarily results in a lower ice onset.

As before (see Section 4.1.1), increasing the geometrical radius $r_a$ of the dry aerosol population also results in a decreased ice onset humidity again caused by the presence of larger solution particles Since the decrease of the ice onsets is qualitatively similar as in the case of increasing $\sigma_a$, these results are not shown here.

## 4.2 The Effect of the Saturation Vapor Pressure

Inspecting again the individual panels in the Figures 6 and 7, the predicted number of nucleated ice crystals appears as mostly unaffected by the choice of the parameterization of the saturation vapor pressure. Although not shown, this is also observed in the simulation results employing the computation of the ice water activity $a_w^i$ as the quotient $\frac{p_{i,\mathrm{sat}}}{p_{\mathrm{sat}}}$ (see Equation (3)) instead of the parameterization given by Koop et al. (2000). This conclusion is consistent with the conclusion presented in Spichtinger et al. (2021).

A larger influence of the choice of the formulation of $p_{\mathrm{sat}}$ is observed by considering the ice onset humidities in Figures 9 and 10. Using Nachbar's parameterization (upper right panels) results in slightly higher ice onset humidities compared to, e.g., using Murphy & Koop's parameterization (upper left panels), in particular at cold temperatures. This may be explained as follows. The saturation water vapor parameterizations satisfy the estimate

$$p_{\mathrm{sat,MK}}(T) < p_{\mathrm{sat,Nach}}(T) \tag{15}$$

for a given temperature $T$, in particular at cold temperatures, see the inequality (8) and Figure 3. For a given humidity $p_v$, expressed as the partial pressure of water vapor, the corresponding saturation ratio is given by $S = \frac{p_v}{p_{\mathrm{sat}}(T)}$. Inequality (15) translates into the inequality $S_{\mathrm{MK}} > S_{\mathrm{Nach}}$ for the corresponding saturation ratios. According to Köhler theory, the latter inequality implies the relation $r_{\mathrm{MK}} > r_{\mathrm{Nach}}$ for the radius of the resulting "grown" solution particles. Thus the computation using the smaller value $p_{\mathrm{sat,MK}}$ predicts a larger grown solution particle, which admits a higher freezing probability, see Equation (1). In summary, employing a parameterization of the water saturation vapor pressure, which yields smaller values in comparison to a second parameterization, results in an earlier freezing. This can also be seen in the lower panels of Figure 9, where the results using $p_{\mathrm{sat}}$ according to Tabazadeh (lower left) and Luo (lower right) are shown. Both parameterizations of $p_{\mathrm{sat}}$ admit values that are in between those of $p_{\mathrm{sat,MK}}$ and $p_{\mathrm{sat,Nach}}$, resulting in ice onset humidities which are also in between.

Note further, the ice onset humidities within the individual panels in Figures 9 and 10 using the equilibrium assumption for the water activity (blue dots) do not change. The reason is found in the assumed formulation of the ice activity $a_w^i$. More precisely, the implementation is often based on the informations found in Koop et al. (2000), hence uses $a_w = S_i a_w^i$ and computes the current ice saturation ratio from the current relative humidity $\mathrm{RH}_w$ by evaluating

$$a_w = S_i a_w^i = \mathrm{RH}_w \frac{p_{\mathrm{sat}}}{p_{i,\mathrm{sat}}} a_w^i = \frac{p_v}{p_{\mathrm{sat}}} \frac{p_{\mathrm{sat}}}{p_{i,\mathrm{sat}}} a_w^i = \frac{p_v}{p_{i,\mathrm{sat}}} a_w^i, \tag{16}$$

where $p_v$ is the partial vapor pressure. Evidently, the expression (16) is independent from the chosen formulation of $p_{\text{sat}}$. Choosing the parameterization for $a_w^i$ given in Koop et al. (2000, their equations 1 and 2), the ice water activity is a function of temperature only, hence (16) is truly independent of $p_{\text{sat}}$. In contrast, choosing to compute the ice activity $a_w^i$ according to the quotient (3) restores the dependency of (16) from $p_{\text{sat}}$. To illustrate this fact, Figure 11 comprises the ice onsets as in Figure 9, now computed using $a_w^i = \frac{p_{i,\text{sat}}}{p_{\text{sat}}}$, see Equation (3). In this figure, the ice onsets based on the equilibrium assumption (blue dots) are clearly not constant anymore and the largest variability is seen at cold temperatures where the various formulations of the saturation vapor pressure $p_{\text{sat}}$ differ the most.

## 4.3 The Effect of the Water Activity

Inspecting Figures 6 and 7 for the number of nucleated ice crystals, the influence of the method to compute the water activity has hardly any effect at low vertical velocities, i.e. for about $w \leq 0.1\,\mathrm{m\,s^{-1}}$, except for slightly increased ice crystal numbers at the lowest vertical velocities and using the equilibrium assumption at the lowest vertical velocities together with small aerosol sizes and a low geometrical width. This is attributable to the low ice onset seen in Figure 9 (blue dots), where the freezing starts early, but the aerosol particles and hence also the nucleated ice crystals are very small, resulting in only a weak sink for the water vapor due to their diffusional growth.

For large vertical velocities (e.g. larger than $1\,\mathrm{m\,s^{-1}}$), the equilibrium-based water activity (circle markers in Figures 6 and 7) appears to produce a higher number of nucleated ice crystals compared to Luo's or Carslaw's parameterization. Apart from that, no systematic influence of the formulation of the water activity on the number of nucleated ice crystals is ascertainable.

The influence of the formulation of the water activity on the ice onset humidity is more pronounced, see the Figures 9 and 10. The ice onset humidities as computed with the equilibrium-based formula (4) are nicely aligned with the "Koop line" (blue dots and dashed blue line), as could be anticipated from the discussion in Section 4.2. Apart from the generally valid influence of the geometrical width $\sigma_a$ on the ice onset, i.e. a larger width of the aerosol size distribution decreases the ice onset, there appears to be no further influence on the ice onsets as obtained by assuming equilibrium.

The ice onset humidities are much more influenced by using either Luo's (red dots) or Carslaw's (green dots) method to compute the water activity. As an example, consider the ice onset humidities in Figure 9. The green dots indicate a higher ice onset compared to the red dots at cold temperatures, while their order is reversed at warm temperatures. This observation may be explained by recalling Figure 4 where the ratio

$$a_{w,\text{Luo,z}}(x,T)/a_{w,\text{Car}}(x,T) \tag{17}$$

is shown.

At cold temperatures, the ratio (17) is mostly larger than unity (red colors in Figure 4), hence an earlier freezing with Luo's parameterization is expected, implying the red dots are below the green dots in Figure 9. At warmer temperatures, the ratio is mostly smaller than unity (blue colors in Figure 4) and an earlier freezing is expected with Carslaw's parameterization, implying the green dots are below red dots in Figure 9.

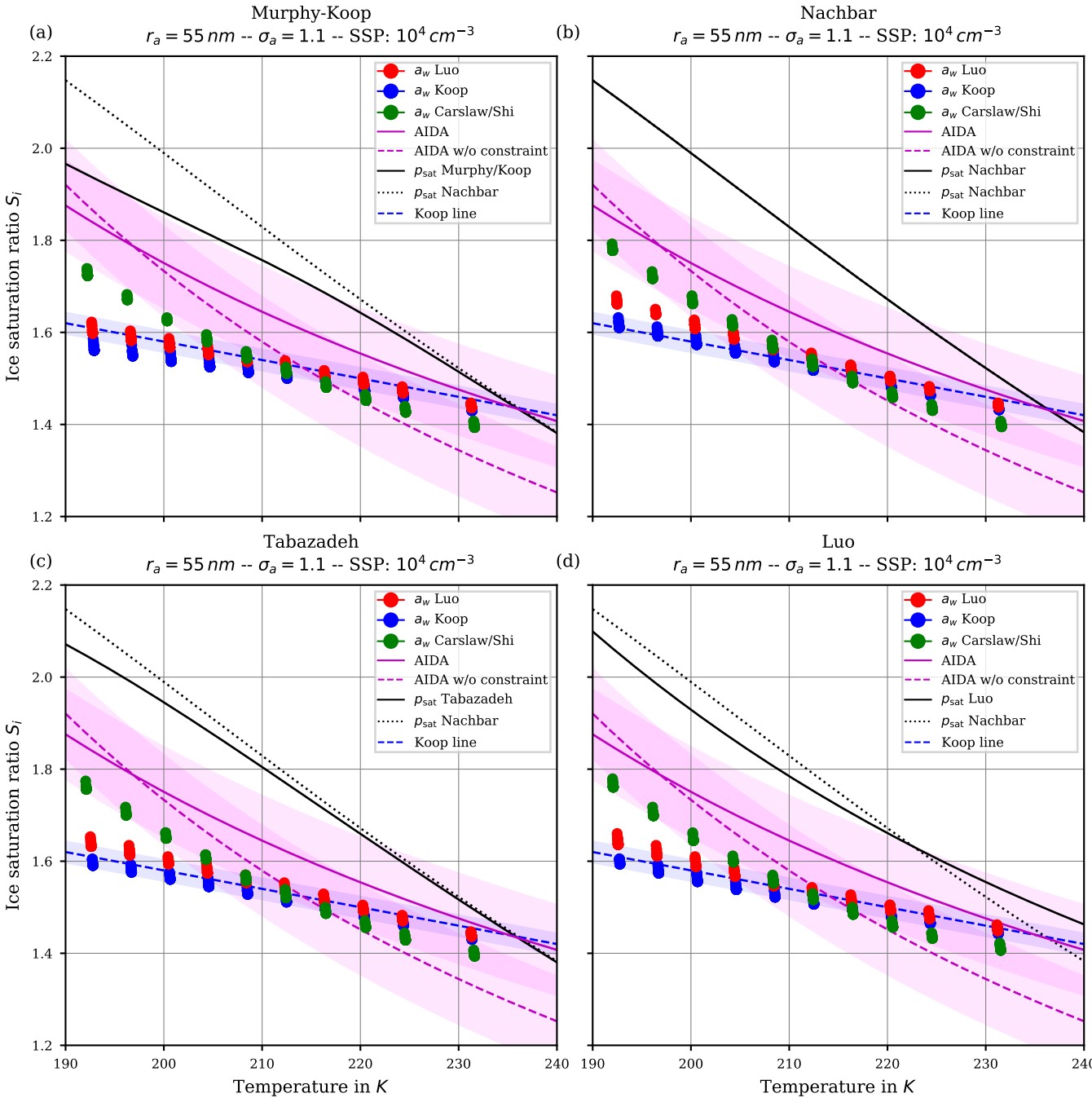

**Figure 11.** Ice onset as a function of temperature for the same scenarios as in Figure 9 with the choice $\sigma_a = 1.1$ for the dry aerosol lognormal size distribution and the ice activity is evaluated by the quotient $\frac{p_{i,\text{sat}}}{p_{\text{sat}}}$, see Equ. (3). The format of the panels is as in Figure 9.

Arguably, the ratio (17) shown in Figure 4 is neither exclusively larger or smaller than unity for a given temperature. As an example, at $195\,\mathrm{K}$ the ratio is smaller than unity for about $0 \leq x \leq 0.15$, larger than unity for about $0.15 \leq x \leq 0.5$ and again smaller than unity for larger mass-fractions. Consequently, the earlier freezing onset for Luo's parameterization at this temperature seen in Figure 9 implies that the composition of the solution particles is within the interval $0.15 \leq x \leq 0.5$.

### 4.4 Fixing Carslaw's Method

As already indicated in Section 3.3, the CLaMS-Ice model admits two locations within the code where the water activity is evaluated. Consequently, one could not only change the method for computing the water activity within the nucleation routine, but also within the routine computing the Köhler equilibrium for the aerosol particles. All the preceding results are based on computations, where Luo's method was employed within the Köhler equilibrium routine. Therefore, we now explore how the results do change by fixing Carslaw's method within that routine.

By inspecting the output we conclude that the number of nucleated ice crystals are largely unaffected by this change, hence these results are not shown here.

The left column in Figure 12 shows the ice onset humidities for a dry aerosol log-normal size distribution with geometric radius $r_a = 55\,\mathrm{nm}$, geometric width $\sigma_a = 1.1$ and the choices $p_{\mathrm{sat,MK}}$ (upper panel), $p_{\mathrm{sat,Nach}}$ (middle panel), $p_{\mathrm{sat,Tab}}$ (lower panel) for the saturation vapor pressure of water. Comparing these with the respective panels from Figure 9, the ice onset humidities are now much lower. However, observe that the general qualitative pattern is still valid, i.e. the red dots are below the green dots at cold temperatures and vice-versa at high temperatures.

In any case, it is worth to comment on the "consistent" model configurations, i.e. the configurations where the same method for the water activity is used at both model code locations, where it is needed. In case of Figure 9, Luo's method was fixed within the Köhler routine, hence the consistent configuration is represented by the red dots. For Figure 12 the consistent configuration is given by the green dots, i.e. Carslaw's method. Notably, the ice onset humidities predicted by the consistent configurations (i.e. red dots in Figure 9 and green dots in Figure 12) are remarkably similar. This is mainly attributed to $f_{1 \mapsto 2}$ being equal to the identity in case of a consistent configuration, cf. the discussion in Section 3.3.

For completeness, the right column in Figure 12 shows the same results as the left column, but with the ice activity $a_w^i$ computed using the quotient $\frac{p_{i,\mathrm{sat}}}{p_{\mathrm{sat}}}$, cf. Equation (3).

It remains to explain the drop in the ice onset humidities caused by the change of the method to compute the water activity within the Köhler equilibrium routine. Since this was the only change made between the results in Figure 9 and the left column in Figure 12, the reason is to be found by comparing the influence of the two formulations of the water activity on the Köhler equilibrium.

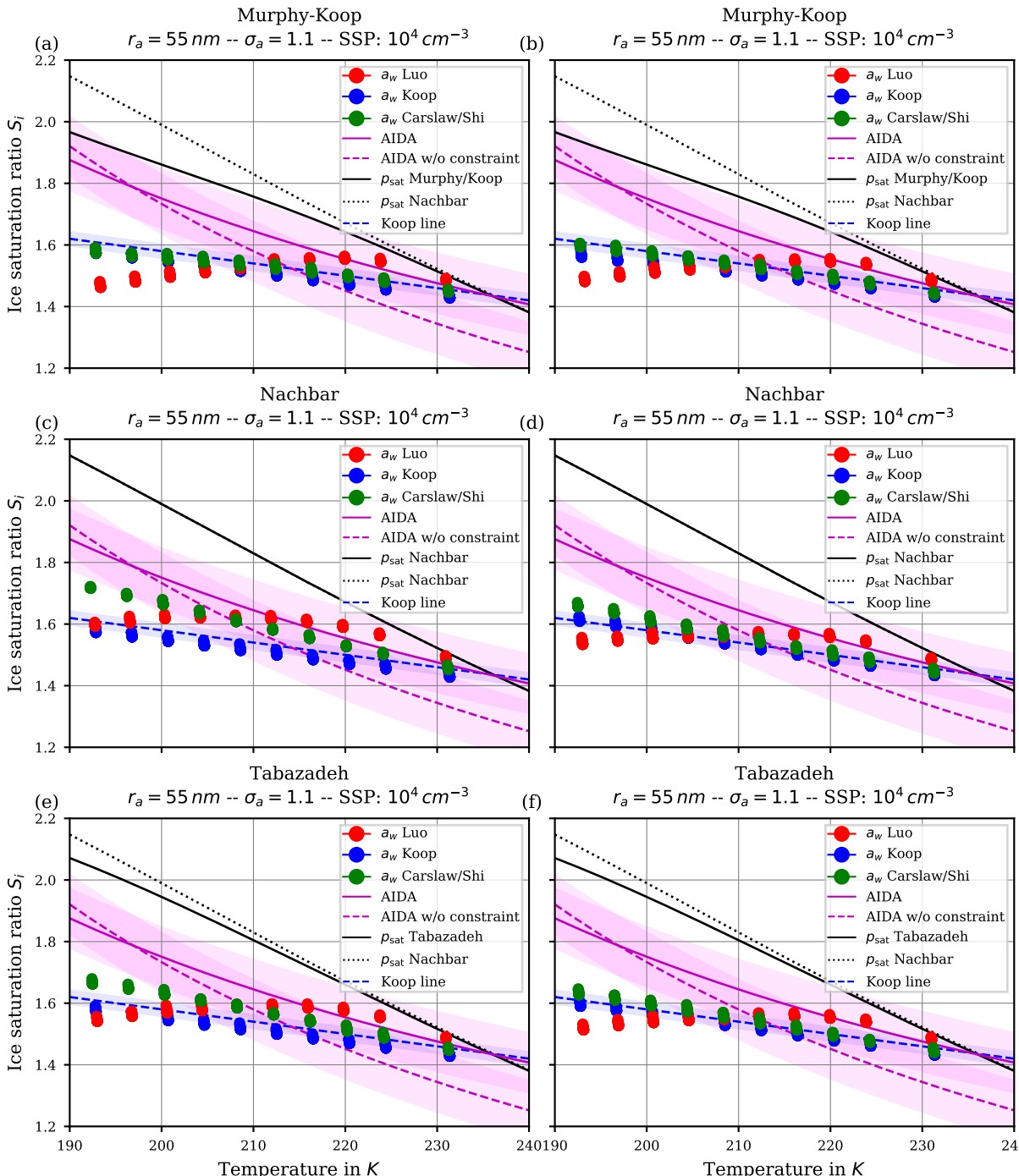

**Figure 12.** Left column: Ice onset humidities as computed by fixing Carslaw's method within the routine computing the Köhler equilibrium and using $a_w^i$ as in Koop et al. (2000). Right column: Same as for the left column but $a_w^i$ is computed by using the quotient $\frac{p_{i,\text{sat}}}{p_{\text{sat}}}$, see Equ. (3). The format of the panels is as in Figure 9.

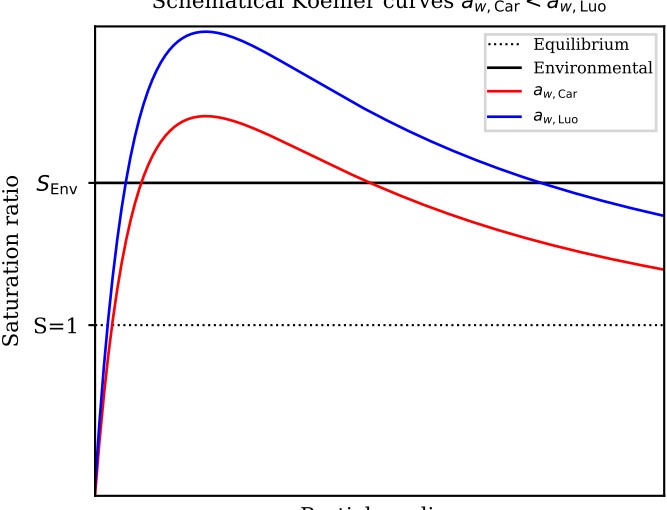

**Figure 13.** Schematic Köhler curves (red and blue curve) for fixed environmental temperature and chemical composition of the solution particle but computed using different representations of the water activity. Here, the assumption $a_{w,\mathrm{Car}} < a_{w,\mathrm{Luo}}$ is adopted; the reversed estimate is obtained by interchanging the role of the red and the blue curve. The solid black line indicates a fixed environmental humidity, represented by the environmental saturation ratio $S_{\mathrm{Env}}$. The dotted black line refers to equilibrium.

Köhler theory (see, e.g., Wang, 2013; Lamb and Verlinde, 2011; Pruppacher and Klett, 2010; Petters and Kreidenweis, 2007) predicts the equilibrium saturation vapor pressure $p_{\mathrm{sat}}(x, T, r)$ over a spherical solution particle with radius $r$, temperature $T$, and amount of chemical substance $x$ (in this case sulfuric acid) as

$$p_{\mathrm{sat}}(x, T, r) = p_{\mathrm{sat}}(T) a_w(x, T) \exp\left( \frac{2\sigma}{R_v T \rho r} \right), \tag{18}$$

where $\sigma, \rho$, and $R_v$ denote the surface tension, density, and the gas constant of water vapor, respectively. In CLaMS-Ice, the parameterizations for the surface tension $\sigma$ and the density $\rho$ are taken from Tabazadeh et al. (2000)[4] and Oca et al. (2018), respectively. Note, since the water activity depends on the amount $x$ of sulfuric acid within the particle, it is also a function of the particle radius $r$ with $a_w \sim \frac{1}{r^3}$ (Wang, 2013; Petters and Kreidenweis, 2007). The exponential factor in (18) encodes the Kelvin effect. Figure 13 shows two schematical Köhler curves, i.e. the equilibrium saturation ratio, as a function of solution particle radius (red and blue curves). In the following, assume a fixed temperature $T$ and a fixed environmental saturation ratio $S_{\mathrm{Env}}$, the latter is indicated by the solid black curve in Figure 13. Moreover, assume the relation

$$a_{w,\mathrm{Luo}} > a_{w,\mathrm{Car}} \tag{19}$$

---

[4]The formula in Tabazadeh et al. (2000) is based on the work from Myhre et al. (1998).

holds for the two possible methods to compute a water activity, corresponding to the red colors in the ratio plot in Figure 4. The Köhler curves (red and blue curves) in Figure 13 conform to this assumption. The black curve indicating the environmental humidity $S_{\mathrm{Env}}$ intersects the two Köhler curves in the stable regime, i.e. at radii smaller than the critical radius which corresponds to the maximum of the Köhler curve. Consequently, the radii corresponding to the intersection points satisfy the relation (see Figure 13)

$$r_{\mathrm{Luo}} < r_{\mathrm{Car}}. \tag{20}$$

These radii represent the radii of the equilibrium solution particles. In other words, given the relation (19) for the water activities, the computation of the size of the solution particles using Carslaw's method results in the prediction of larger solution particles, which freeze earlier. This explains the lower ice onset humidities in Figure 12 compared to Figure 9 where Luo's method was used. For larger temperatures, the relation (19) is reversed, resulting in a reversed estimate (20).

## 4.5   Ice nucleation formulations best matching to observations


Next we address the question, which combination of parameterizations presented in the last Sections could be recommended to best represent the homogeneous ice nucleation process in terms of the number of nucleated ice crystals and the ice onset humidity. A natural approach towards answering this question is to compare the computed results with observations. As already indicated, homogeneous ice nucleation experiments performed at the AIDA cloud chamber are used for that purpose, see
Schneider et al. (2021).

### 4.5.1   Ice onset humidity

Together with the results of the ice onset sensitivity simulations, Figures 9 - 12 in Section 4.5 include fit curves of the ice onset humidities observed at the AIDA cloud chamber (magenta curves, again shown in Figure 14). For completeness, a magenta-shaded region of width 0.1 towards smaller and larger values is included to account for measurement uncertainties
and variabilities within the experiments. Much more details about the measurements and their uncertainties are described in Schneider et al. (2021). Note again, the dashed magenta curve is a fit through the observational data whereas the solid magenta curve additionally takes a constraint into account, i.e. the fit curve should intersect the water saturation line at $235\,\mathrm{K}$. As outlined by Schneider et al. (2021), the ice onsets observed at the AIDA chamber are clearly above the homogeneous freezing thresholds proposed by Koop et al. (2000); see the blue dashed line in Figure 14 and compare, e.g. in Figure 9, the blue dots
(representing Koop et al. (2000)) with the solid and dotted magenta lines.

    A modelled ice onset humidity is considered to be close to the observations, if the data point is within the magenta-shaded region. Judging from the aforementioned figures showing the atmospherically relevant (geometric) dry particle radius $r_a = 55\,\mathrm{nm}$, the computed ice onset humidities for both "consistent" simulations[5] are similar and match best with the AIDA ice

---

[5]"consistent" simulations: the water activity parameterization of either Luo or Carslaw is used for both ice nucleation and aerosol growth (Köhler equilibrium); Luo: red dots in Figure 9, Carslaw: green dots in Figure 12; see Section 4.4

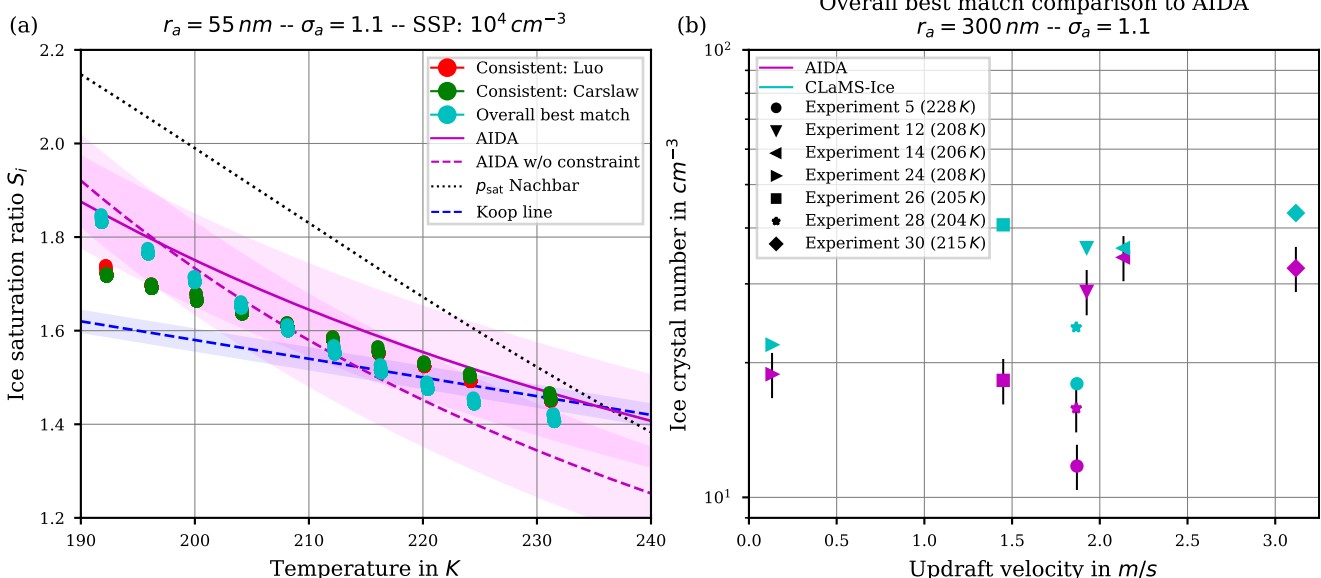

**Figure 14.** (a) Ice onset humidities as a function of temperature and (b) number of nucleated ice crystals versus vertical updraft. **Magenta colours**: observations at the AIDA cloud chamber ((a) dotted line: fit line through observational results for sulfuric acid solution particles; solid line: fit line crossing water saturation at $235\,\mathrm{K}$; shaded regions: uncertainties of $\pm0.1$ around fit lines; (b) symbols: experiments performed during the campaign TROPIC02, see Schneider et al., 2021), together with the observed ice onset temperature per experiment. **Cyan symbols**: CLaMS-Ice simulations "overall best match" with AIDA observations, achieved by combining the water vapor saturation parameterization $p_{\mathrm{sat}}$ from Nachbar et al. (2019) with Carslaw's water activity $a_w$ for ice nucleation and Luo's $a_w$ for aerosol growth (Köhler equilibrium). **Red and green dots**: CLaMS-Ice simulations "consistent" setups, i.e. Nachbar's $p_{\mathrm{sat}}$ and either Luo's (red dots) or Carslaw's (green dots) $a_w$ for both ice nucleation and aerosol growth.

onsets by using Nachbar's parameterization for the water saturation pressure (i.e., $p_{\mathrm{sat,Nach}}$). Figure 14 is intended to provide
a summary of the findings and shows these data points again as the red and green dots. The simulations indicate, the AIDA observations are best resimulated by using the saturation vapor pressure of water that yields highest values at low temperatures.

     However, it should be noted that an even better agreement of the simulation results with the AIDA observations is possible by abandoning the requirement to use one of the "consistent" combinations. As an example, inspecting Figure 9, the overall best match with the observational data is achieved (see the upper right panel, green dots) by using Carslaw's method for the
water activity in the nucleation routine and Luo's method in the Köhler equilibrium routine, in combination with Nachbar's formulation of the saturation vapor pressure. The same datapoints for this "overall best match" are reproduced in Figure 14 as the cyan dots.

     In passing, we note that the unconstrained AIDA fit, i.e. the dashed magenta curve, indicate values of the ice onset humidities well below the Koop line (dashed blue curve) at temperatures above about $215\,\mathrm{K}$. Although it is not clear at the moment if this

deviation could result from measurement artifacts, the modelled "overall best match" ice onsets indicate the same qualitative behaviour at warm temperatures.

### 4.5.2 Number of nucleated ice crystals

The number of homogeneously nucleated ice crystals within our simulation results agree well with the expected number concentrations from the reference study by Kärcher and Lohmann (2002), see, e.g., Figure 6. Moreover, as long as the mean size
and width of the size distribution of the aqueous solution particles from the AIDA experiments are known, running the CLaMS-Ice model along the measured input trajectories yields numbers of homogeneously nucleated ice crystals that agree well with the observed number of ice crystals. An example is shown in Figure 14b, where the observed ice crystal number concentration at the AIDA chamber are compared with the number concentrations as predicted by CLaMS-Ice using the "overall best match" model configuration. The agreement between the observed and simulated ice crystal number concentrations is largely independent
of the employed parameterization of the water vapor saturation pressure and the formulation of the water activity (not shown here). This conclusion is consistent with the conclusion described in the previous sections, i.e. the number of ice crystals is less sensitive to the choices of the parameterizations in comparison to the ice onset humidity.

## 5   Summary and Conclusions

Homogeneous ice nucleation plays an important role in the formation of cirrus clouds. It is well understood and described by
the water activity based approach introduced by Koop et al. (2000), which is nowadays implemented in most models simulating cirrus ice clouds. However, the results in Koop et al. (2000) are based on specific formulations for the water vapor saturation pressure $p_{\mathrm{sat}}$ and the water activity $a_w$, but there are several descriptions available in the literature for both parameters. This study reports on numerical simulations of homogeneous ice nucleation using the CLaMS-Ice model to study the sensitivity to four different formulations for $p_{\mathrm{sat}}$ (Luo et al., 1995; Tabazadeh et al., 1997; Murphy and Koop, 2005; Nachbar et al., 2019)
and three for $a_w$ (Luo et al., 1995; Koop et al., 2000; Shi et al., 2001, where Shi's $a_w$ is based on the formulation by Carslaw et al. (1995)).

   CLaMS-Ice is a trajectory-based model, i.e. it computes ice microphysics along given air parcel trajectories by using the recent two-moment bulk ice microphysics scheme by Spichtinger and Gierens (2009). For the present study, most air parcel trajectories describe an artificial adiabatically ascending air parcel with constant vertical velocity as in Kärcher and Lohmann
(2002), while some trajectories are based on ice cloud experiments at the AIDA cloud chamber. The artificial trajectories are chosen to be able to extract the pure microphysical effects on the ice nucleation when changing the formulations of $p_{\mathrm{sat}}$ and $a_w$; the AIDA cloud chamber trajectories are used to compare simulations with observations. The results are presented in two different phase spaces: (i) the vertical velocity - ice number concentration phase space that resembles the format of the reference study by Kärcher and Lohmann (2002), and (ii) the temperature - ice onset humidity phase space introduced by Koop
et al. (2000).

Several characteristics of these phase-space plots are discussed, for example the impact of changes within the size distribution of the dry aerosol particles on the number of homogeneously nucleated ice crystals. In general, larger particles freeze earlier, hence both an increase in the mean size of the aerosol particles or a broader size distribution lead to an earlier ice onset, i.e. fewer ice crystals appear at warmer temperatures and lower ice supersaturations (see Section 4.1.1). Good agreement between the ice crystal numbers simulated with CLaMS-Ice and measurements at the AIDA cloud chamber is found when using the observed parameters of the aerosol particle size distribution (see Figure 14b), in line with Jensen et al. (1998), who compared simulated ice crystal numbers with aircraft measurements in wave-clouds.

Of special importance for this study is, however, that both the choice of the parameterization of the water saturation and the method to compute the water activity have only a minor impact on the number of nucleated ice crystals (see Section 4.1.1). This finding is consistent with conclusions of Spichtinger et al. (2021) and is particularly important for the use of two moment schemes in large-scale and global models.

The ice onset humidities are, however, much more sensitive to the choices of the water saturation and water activity formulations (see Section 4.1.2). Requiring a "consistent" model setup, i.e. using the same method for the water activity when computing ice nucleation rate coefficients and the growth of aerosol particles (Köhler equilibrium), the best match with ice onset humidities observed at the AIDA cloud chamber is obtained by using the methods of either Luo or Carslaw/Shi together with Nachbar's formulation for the water saturation (see the red and green dots in the summary Figure 14a in comparison to the magenta AIDA lines). Relaxing the requirement of internal model consistency, an even better agreement is possible by still using Nachbar's formulation for the water saturation but employing a mixed strategy for the formulation of the water activity, i.e. Carslaw/Shi's method within the computation of the ice nucleation rate coefficient and Luo's method within the computation of aerosol growth. This "overall best match" is represented as the cyan dots in Figure 14a.

Note here that, unlike Koop's approach to homogeneous ice nucleation which is independent of the aerosol particle chemistry, the recommended combination of the parameterizations is specialized to aqueous sulfuric acid particles, since the various methods for the water activity are formulated using parameterizations based on sulfuric acid. Consequently, it may give less accurate predictions if the solution particles contain significant amounts of other substances. However, since sulfuric acid is believed to be the most prominent ingredient of solution particles at cirrus levels employing the two moment scheme with this parameterization in a large scale model should still provide reasonable results.

Altogether, the formulation of the water activity and water saturation significantly affects the ice onset humidity, which can also be interpreted as the time (or temperature) of the ice onset within an adiabatically ascending air parcel. A lower/higher ice onset humidity thus means that the cirrus ice clouds would appear earlier/later at lower/higher temperatures, affecting the coverage of the sky with cirrus clouds, hence inducing an effect on their radiative properties. In particular at cold temperatures, huge differences between the various formulations are observed, so modelling studies in this temperature range are most affected by the choice of the water activity and water saturation. Such studies include simulations of cirrus clouds in the tropical transition layer (TTL), where the homogeneous ice onset humiditiy is of particular importance because it determines not only the appearance of cirrus clouds, but also the maximum clear sky and in-cloud humidity and with this the amount of water available for further transport into the stratosphere, which in turn has an impact on the Earth's climate (Riese et al.,

2012). However, we note that our choice of air parcel temperatures for the simulations is inspired by the choices in Kärcher and Lohmann (2002) and does not cover the coldest environmental conditions found in the TTL that may range down to about 180 K.

Krämer et al. (2009, 2020) indicated possible clear sky supersaturations above the homogeneous freezing threshold of Koop et al. (2000) in the TTL at temperatures below about 205 K, pointing to a potentially higher homogeneous ice onset humidity. Enhanced ice onset humidities at these cold temperatures are reported as a general feature from the ice nucleation experiments at the AIDA cloud chamber by Schneider et al. (2021) (see Figure 14a, magenta lines: new fit line for the homogeneous ice onset humidity based on the AIDA measurements). From our CLaMS-Ice sensitivity simulations with respect to different water saturation and water activity formulations, we here provide a model framework that is able to reproduce the new, enhanced freezing line. In future studies, we will explore the effect of a higher homogeneous freezing threshold on the TTL cirrus coverage and also the corresponding transport of water vapor into the stratosphere.

## Appendix A: Parameterizations

This appendix serves to summarize formulas of the parameterizations used in this study. In the following, exclusively SI units are used, i.e. temperature $T$ is in kelvin and pressure is in pascals.

### A1  Saturation Vapor Pressure over Ice

The parameterization of the saturation vapor pressure over a flat ice surface is taken from Murphy and Koop (2005) and reads

$$p_{i,\text{sat}}(T) = \exp\left(9.550426 - \frac{5723.265}{T} + 3.53068\log(T) - 0.00728332T\right) \tag{A1}$$

where $\log$ denotes the logarithm with base $e$. This formula is valid for $T > 110\,\text{K}$.

### A2  Saturation Vapor Pressure over Liquid Water

The formulation of the saturation vapor pressure over liquid water from Murphy and Koop (2005) is given by

$$\log\left(p_{\text{sat,MK}}(T)\right) = 54.842763 - \frac{6763.22}{T} - 4.210\log(T) + 0.000367T + \tanh\left(0.0415(T - 218.8)\right)$$
$$\cdot \left[53.878 - \frac{1331.22}{T} - 9.44523\log(T) + 0.014025T\right] \tag{A2}$$

which is valid for $123\,\text{K} < T < 332\,\text{K}$.

The formulation by Tabazadeh et al. (1997) reads

$$\frac{1}{100}\log\left(p_{\text{sat,Tab}}(T)\right) = 18.452406985 - \frac{3505.1578807}{T} - \frac{330918.55082}{T^2} + \frac{12725068.262}{T^3} \tag{A3}$$

and is stated to hold for the temperature range $185\,\text{K} \leq T \leq 260\,\text{K}$.

The more recent parameterization by Nachbar et al. (2019) is given by

$$\log\left(p_{\text{sat,Nach}}(T)\right) = 74.8727 - \frac{7167.40548}{T} - 7.77107\log(T) + 0.00505T \tag{A4}$$

for temperatures $T > 200\,\mathrm{K}$.

Note, Luo's parameterization $p_{\mathrm{sat,Luo}}$ is given in subsection A3, since it is obtained by substituting $x = 0$.

## A3   Saturation Vapor Pressure over Sulfuric Acid Solution

The work by Luo et al. (1995) provides a parameterization of the saturation vapor pressure over a binary $H_2SO_4/H_2O$ solution, given by

$$\frac{1}{100}\log(p_{\mathrm{sat,Luo}}(x,T)) = 23.306 - 5.3465x + 12w_h x - 8.19w_h^2 x + \frac{-5814 + 928.9x - 1876.7w_h x}{T} \tag{A5}$$

with the mass-fraction $x$ of sulfuric acid, $w_h = 1.4408x$, and the temperature $T$.

## A4   Direct Parameterization of Water Activity

The study by Shi et al. (2001) provides a formula which allows to directly evaluate the water activity $a_w$ for a binary $H_2SO_4/H_2O$ solution and is based on the work from Carslaw et al. (1995). This formula reads

$$\log(a_{w,\mathrm{Car}}(x,T)) = \left[-69.775X - 18253.7X^2 + 31072.2X^3 - 25668.8X^4\right] \cdot \left[\frac{1}{T} - \frac{26.9033}{T^2}\right] \tag{A6}$$

with the mole-fraction $X$. The latter is connected to the mass-fraction $x$ by the relation

$$X = \frac{x}{x + (1-x)\frac{98.078}{18.015265}}. \tag{A7}$$

## Appendix B:  Application of the water activity formulas in the model

When applying one of the possibilities (ii) or (iii) from Section 2.3.1 to compute the water activity $a_w$ in models the formulations needs to be adjusted to the chosen water vapor saturation pressure $p_{\mathrm{sat}}$ (cf. Section 2.2). The reason is, both of the formulations (10), (11) are constructed using a specific choice for $p_{\mathrm{sat}}$. Thus, the formulations (10), (11) need to be multiplied by the factor

$$\frac{p_{\mathrm{sat,spec}}(T)}{p_{\mathrm{sat}}(T)}, \tag{B1}$$

where $p_{\mathrm{sat,spec}}(T)$ denotes the choice of parameterization of the saturation vapor pressure used in the development of the formulation of $a_w$ and $p_{\mathrm{sat}}(T)$ is the desired formulation of $p_{\mathrm{sat}}$. Thus, for any desired target formulation $p_{\mathrm{sat}}$, the model computations are done by applying the expressions

$$a_{w,\mathrm{Luo}}(x,T) \cdot \frac{p_{\mathrm{sol,Luo}}(0,T)}{p_{\mathrm{sat}}(T)} \tag{B2}$$

and

$$a_{w,\mathrm{Car}}(x,T) \cdot \frac{p_{\mathrm{sat,Car}}(T)}{p_{\mathrm{sat}}(T)} \tag{B3}$$

instead of (10) and (11), respectively. Formulation $p_{\mathrm{sat,Car}}(T)$ is based on equation 19 and table 1 of Carslaw et al. (1995). Noteworthy, this formulation is very similar to $p_{\mathrm{sat,Tab}}$.

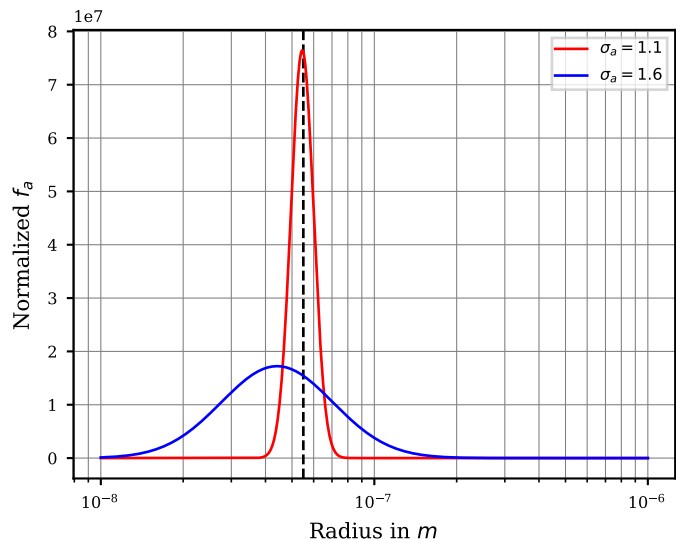

**Figure C1.** Two lognormal size distributions based on (13) with $N_a = 1$. The size distributions share the value $r_a = 55\,\text{nm}$ for the geometric mean radius (indicated by the vertical dashed line) but feature different choices for the geometric width $\sigma_a = 1.1$ (red curve) and $\sigma_a = 1.6$ (blue curve).

## Appendix C:  How expressive is the geometric radius for broad aerosol size distributions?

According to (1), the probability of freezing of a given aerosol particle is higher the larger the particle is. In this appendix, we illustrate how expressive the geometric mean radius $r_a$ is if the aerosol size distribution follows a lognormal size distribution (13). Consider the (normalized) size distributions in Figure C1 with geometric mean radius $r_a = 55\,\text{nm}$ and the two choices $\sigma_a = 1.1$ (red curve), $\sigma_a = 1.6$ (blue curve), representing a narrow and a broad size distribution. For the narrow size distribution, the majority of aerosol particles have sizes that are comparable to the geometric mean radius $r_a$. Consequently, the freezing properties of the majority of the aerosol particles are similar and the mean radius provides a good description of the whole particle population. In contrast, the broad size distribution features many aerosol particles which are significantly larger than the size indicated by $r_a$. These large aerosol particles freeze much earlier than the smaller particles, hence a description of the freezing behaviour based on $r_a$ is less expressive.

## Appendix D:  Example simulation with small timestep

In order to illustrate the effect of a reduced timestep and elucidate the model artifact, the simulation using $r_a = 55\,\text{nm}$ and $\sigma_a = 1.1$ (cf. Figure 6) is redone with the fixed timestep $5\,\text{ms}$ as in Kärcher and Lohmann (2002). The resulting number concentrations of nucleated ice crystals are shown in Figure D1. The overestimated number concentrations at high vertical velocities and warm temperatures are reduced and agree better with the reference results compared to the results using larger

timesteps as shown in Figure 6. Note, the range of vertical velocities is restricted to large vertical velocities to more clearly show the reduction of the model artifact.

*Author contributions.* MB adapted the CLaMS-Ice model with the help of CR, MK and JUG. MB performed the numerical simulations. MB, MK and all co-authors contributed to the analysis and interpretation of the simulation results and the AIDA observational data, drafting and approval of the manuscript. The observational studies at the AIDA cloud chamber were conducted by JS, TS, supervised by OM.

*Competing interests.* The authors declare that they have no conflict of interest.

*Acknowledgements.* We thank Daniel Knopf for his very valuable comments on the manuscript. In addition, we thank two anonymous reviewers for their comments and suggestions which led to a significantly improved manuscript. Martina Krämer and Manuel Baumgartner acknowledge support from the JGU Mainz during Martina Krämer's fellowship by the Gutenberg Research Council (GFK fellowship). Manuel Baumgartner also acknowledges support by the Deutsche Forschungsgemeinschaft (DFG) within the Transregional Collaborative
Research Centre TRR165 Waves to Weather, (www.wavestoweather.de), project Z2. Peter Spichtinger acknowledges support by the DFG through the research unit Multiscale Dynamics of Gravity Waves (MS-GWaves) and through grant SP 1163/5-2. Also, the support by the DFG through the project KR2957/4-1 (TropiC, German part of the PIRE project) is greatly appreciated. Finally, we thank the developers of *GNU parallel* for their incredible useful parallelization tool (Tange, 2011), which was used in this study.

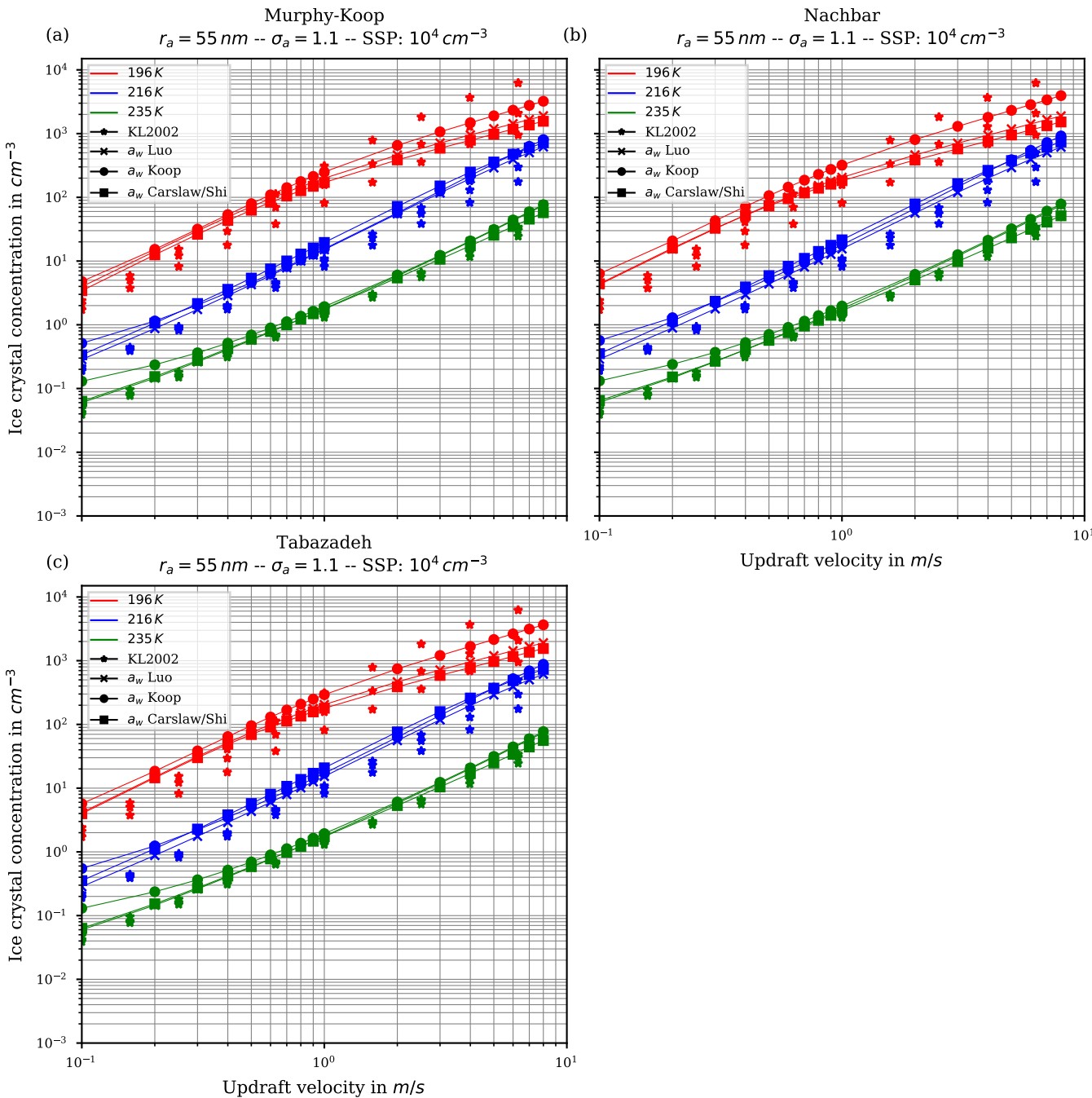

**Figure D1.** Same as in Figure 6, but with fixed timestep 5 ms. Note, the range of vertical velocities is restricted to large vertical velocities to more clearly show the reduction of the model artifact.

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
