# Peer review of "New investigations on homogeneous ice nucleation: the effects of water activity and water saturation formulations"

_Atmospheric Chemistry and Physics, 2021_

## Referee Comment (RC1)

**Review of "New investigations on homogeneous ice nucleation: the effects of water activity and water saturation formulations" by Baumgartner et al.**

**Verdict**

I recommend that the paper be published subject to major revisions.

**General comments**

At the start of the paper, it needs to be clarified that there are two broad types of homogeneous freezing: that of solution droplet aerosols (CCN) and that of cloud-liquid/rain. It also needs to be clarified that the paper is only about the former.

An intellectual weakness of the paper is that there seems to be a rather ad hoc trial of various combinations of water activity and vapour pressure schemes, with a recommendation of an optimal combination without clear arguments independently for why these schemes are best. So, the reader may infer that there is a likelihood of a balance of compensating biases which could be unsettled if a different chemical species of aerosol (not sulphate) or different thermodynamic conditions were simulated.

Need to argue more about whether the Nachbar scheme is more accurate than the Murphy and Koop scheme for vapour pressure. Yes, the Nachbar scheme is from recognition of a new phase of water. But what do the authors think about the likelihood of this assumption being correct ? What new information has come to light that supports this assumption ?

It would be good to comment on AIDA measurements by Mangold et al. (2005) presented at EGU about *lower* humidities for freezing of ammonium sulphate than predicted by Koop. Was this an anomalous observation perhaps ?

For modellers who use the Koop scheme, we freeze CCN aerosols at temperature- and size-dependent critical saturation ratios. How should we modify our schemes ? Modellers have typically simply created a lookup table of these critical ratios from the published plot of Koop et al. (2000):

[Figure]

Can a simple numerical fix be recommended ? If a line of best fit can be given for the "best fit" of Fig. 15a for r = 0.05 um, do we just shift the line downwards for larger sizes ?

**In summary, this paper represents a significant advance in knowledge. More discussion of the merits of the various schemes (Luo, Nachbar, Carslaw etc) is needed. Also an awareness of the place of homogeneous aerosol freezing among other types of homogeneous freezing needs to be conveyed. Above all, modellers need to be told how to upgrade their lookup tables (e.g. saturation ratio as function of size and temperature) in light of these important results from AIDA.**

**Specific comments**

Line 24: the statement is too narrowly centered on aerosols: "*In contrast, homogeneous nucleation refers to the spontaneous freezing of pre-existing solution particles*". In fact, homogeneous nucleation of ice refers to spontaneous freezing of any drop, whether a solution droplet ('homogeneous aerosol freezing') or a liquid cloud- or rain-drop ('homogeneous freezing of cloud-liquid or rain').

Line 363: It is interesting that, qualitatively, a similar sort of dependency on updraft speed is found with homogeneous freezing of cloud-droplets near -36 degC. With these too, the largest freeze first during ascent and an ascent-dependent fraction of all the cloud-droplets will freeze with the rest evaporating due to the ice particles lowering the humidity. There is preferential freezing of the smaller cloud-droplets. So the number of ice particles initiated increases with updraft speed. Phillips et al. (2007, JAS) parameterized this and showed that it has an order-of-magnitude impact on mesoscale averages of the ice concentration aloft.

Line 573: The paper suggests an optimum combination: "Carslaw's method for the water activity in the nucleation routine and Luo's method in the Köhler equilibrium routine, in combination with Nachbar's formulation of the saturation vapor pressure". But what are the independent reasons for thinking that Carslaw's method is better than alternatives for water activity ? Why is Luo's method better than

alternatives for the Kohler equilibrium routine ?  Why is Nachbar's formulation best ?  Need to provide some independent reasoning or evidence.

Could there be a serendipitous compensation of opposing biases among these three schemes, giving the impression of realism for the wrong reasons ?

**Technical details**

Figure 13:  this has wrong entries in the legend (lines instead of symbols) and multiple panels have identical titles and legends without clarity from the figure caption.

---

## Referee Comment (RC3)

**Review of "New investigations on homogeneous ice nucleation: the effects of water activity and water saturation formulations" by M. Baumgartner et al.**

July 26, 2021

This paper revisits the homogeneous freezing of supercooled liquid aerosol droplets in the cirrus regime. It is motivated by recent observations at the AIDA chamber which apprently disagree with the canonical parameterization provided by Koop et al. (2000). Within Koop's paradigm, the present study examines the sensitivity of the homogeneous nucleation threshold and nucleated ice crystal number to changes in the expressions of liquid water saturation pressure and water activity in solutes.

The manuscript is generally well-written and I appreciate the authors' effort to detail the assumptions behind Koop's homogeneous freezing theory. Nevertheless, I find that major revisions are required before the paper can be considered for publication. I have reservations about part of the approach and several major points (see below) should be taken into account and corrected or clarified.

**Major concern**

In a few simulations, the aerosol equilibrium and nucleation rate calculations employ different, inconsistent formulas for water activity within the aqueous aerosol. This results in an inconsistency, which is acknowledged for example p 29 line 515. One of the inconsistent numerical experiments shows an improved agreement with AIDA ice onset observations (Fig. 15).

In my opinion, this inconsistency alters the original concept of Koop et al. (2000) which not only synthesizes earlier laboratory experiments but also aims at simplicity, taking advantage of thermodynamic constraints. Those include:

$$a_w^i = \frac{p_{i,sat}}{p_{sat}} \tag{1}$$

and Köhler theory for aqueous aerosols in equilibrium, namely:

$$a_w = S_i \, a_w^i \, \exp\left(\frac{-2\sigma}{R_v T \rho \, r}\right) = RH_w \, \exp\left(\frac{-2\sigma}{R_v T \rho \, r}\right) \tag{2}$$

In Equation 2, the resulting equilibrated water activity in the aerosol, $a_w$, depends on the analytical expression of water activity in the solution only through the factor $r$. If the curvature is neglected (i.e. if $r$ is sufficiently large), Equation 2 reduces to the equilibrium formula proposed by Koop et al. (2000):

$$a_w = S_i \, a_w^i \tag{3}$$

in which aerosol water activity is set by environmental conditions only and is virtually independent of aerosol properties. As a consequence, the nucleation probability does not depend (or only marginally, through $r$) on the formula used for water activity in the solute. This is actually noticed (though not explained) by the authors: in consistent set-ups, they obtain similar nucleation thresholds although the formulas used for water activity are different (p 29 lines 518-519).

However, by employing inconsistent expressions for water activity between the aerosol equilibration (Köhler) and the freezing rate calculation, the authors artificially break the link between environmental conditions and the nucleation rate implied in Koop et al. (2000). Namely, instead of having the activity in the solute given by Eq. 3 (as an approximation to Eq. 2), it now follows :

$$a_w = f_{1 \to 2}(S_i \, a_w^i, T) \tag{4}$$

where the function $f_{1 \to 2}$ converts activity from formulation 1 ($a_{w,1}$ used in the Koehler routine) to formulation 2 ($a_{w,2}$ used in the nucleation rate computation) and is defined such that:

$$a_{w,2}(x(r), T) = f_{1 \to 2}(a_{w,1}(x(r), T), T) \tag{5}$$

I believe this implicit replacement of Eq. 2 (or 3) by Eq. 4 (or a similar one with Kelvin correction) is the reason for the significant sensitivity of the nucleation thresholds to water activity formulations in the authors' simulations, not only in Sect. 4.3 but also in Sect. 4.4.

This important point needs to be accounted for and discussed early on by the authors. For clarity, I also would also ask them to drop the "most" when referring to the "most consistent" configurations, since they are "just" consistent. Note that this comment does not affect the conclusions regarding the impact of the formula used for water saturation.

**Other comments**

- I noticed differences between figures 6 and 8 at low vertical velocity where the small time step result exceeds that obtained with the large time step by a factor of about 2. This should be explained or corrected.

- Related to my main comment, I disagree with the authors regarding the impact of aerosol properties (in particular, aerosol radius $r$). At the beginning of the paper, it is argued (as in Koop et al., 2000; Kärcher and Lohmann, 2002) that the dependency on aerosol size is moderate, and cannot explain the differences between the AIDA chamber experiments and simulations based on the classical Koop approach.

Comparing Fig. 10 with Fig. 11 demonstrates that the impact of the aerosol size distribution, albeit present, remains limited.

Then, on several instances, the authors attribute changes in the nucleation threshold to changes in the aerosol radius $r$ associated with Köhler equilibrium: Sect. 2.2, lines 446-460 (which seemingly contradict lines 446-460 where the primary importance of ice activity formulation is recognized) and Sect. 2.4, lines 526-546. As decribed in my main comment, I would guess that the apparent sensitivity to aerosol size is an artifact due to the inconsistent use of different solute water activity formulations.

- The paper rests solely on numerical investigations, whereas the authors have made valuable contributions to the theoretical analysis of homogeneous freezing of aqueous aerosols (e.g. Baumgartner and Spichtinger, 2019). Couldn't some of this analysis be useful here ? It would help explain and synthesize the results (such as the insensitivity of nucleated ice crystal number to the activity formulation).

- Why are the freezing onset simulations limited to the range 190-230 K? The relevant temperature range for cirrus extends down to ∼180-185 K (tropical tropopause layer) as do the experiments of Schneider et al. (2021).

*Presentation*

- The goal of the study is not entirely clear, is it (a) an evaluation paper for the nucleation scheme of CLaMS-ice, or (b) an investigation of the impact of water activity and saturation formulation on homogeneous freezing calculations (as suggested by the title and introduction)? If (b), I would encourage the authors to rephrase part of the text to emphasize that most conclusions apply in general and not only to their specific numerical model. They could remove/shorten some of the many references to the code (l 324, ...).

- The paper could be more concise. For instance, some figures and discussions could be moved to the appendix/ supplementary information. I am in particular referring to Fig 8 (p 18-19) which shows the sensitivity of the model results to the time step and illustrates that, for large vertical velocities, the calculations in Figs 6 and 7 had not converged. I would discuss this in an appendix and only keep figures 6 and 7 with the converged (i.e. 5 ms) time step. Also, some panels in Figs. 6-9 could be moved to the supplementary information, to limit the number of panels in the main body of the paper and make the relevant information more accessible.

- It would be useful to have a table or schematic summarizing where each parameterization/formula is used in the model for a given set of experiments (i.e. the water vapor saturation and water activity parameterizations).

**Specific comments:**

1. p 3 l 60: remove brackets

2. p 3 l 67: is this comment relevant in the introduction ? the point about latent heat release is not discussed further in the text and does not seem essential to me.

3. p 15 Sect 3.4: I would specify already here the aerosol characteristics and water saturation formula used by Kärcher and Lohmann (2002)

4. p 24 l 445: the Spichtinger et al. paper is not yet published. Putting a version on archive would be useful.

5. Fig 10, 11: the equation number of Koop activity is missing in the legend

6. p 29 l 518-519: This is expected I believe (see main comment).

7. p 29, Eq (21): The reference formula/value used for the surface tension $\sigma$ should be specified.

8. p 32, lines 582-583: the authors can reproduce the nucleated number concentration "as long as the mean size and width of the size distribution of the aqueous solution particles from the AIDA experiments are known". How do they fit the dry aerosol radius to match the hydrated observed radius ? How strongly do the presented results depend on this fit? Furthermore, is the distribution of dry aerosols in the previous figures consistent with the aqueous aerosol in the AIDA experiments shown in Fig. 2 of Schneider et al. (2021) ($r \sim 250$ nm) ?

9. Fig. 15: If possible, error bars should be provided for the experimental values.

**References**

Baumgartner, M. and Spichtinger, P.: Homogeneous nucleation from an asymptotic point of view., Theor. Comput. Fluid Dyn., 33, doi:10.1007/s00162-019-00484-0, URL https://doi.org/10.1007/s00162-019-00484-0, 2019.

Kärcher, B. and Lohmann, U.: A Parameterization of Cirrus Cloud Formation: Homogeneous Freezing of Supercooled Aerosols, Journal of Geophysical Research: Atmospheres, 107, AAC 4–1–AAC 4–10, doi:10.1029/2001JD000470, 2002.

Koop, T., Luo, B., Tsias, A., and Peter, T.: Water activity as the determinant for homogeneous ice nucleation in aqueous solutions, Nature, 406, 611–614, doi:10.1038/35020537, 2000.

Schneider, J., Höhler, K., Wagner, R., Saathoff, H., Schnaiter, M., Schorr, T., Steinke, I., Benz, S., Baumgartner, M., Rolf, C., Krämer, M., Leisner, T., and Möhler, O.: High Homogeneous Freezing Onsets of Sulfuric Acid Aerosol at Cirrus Temperatures, Atmospheric Chemistry and Physics Discussions, 2021, 1–30, doi:10.5194/acp-2021-319, URL https://acp.copernicus.org/preprints/acp-2021-319/, 2021.

---

## Author Comment (AC1)

**Response to Reviewer Comments**

Manuel Baumgartner[1,2,a], Christian Rolf[3], Jens-Uwe Grooß[3], Julia Schneider[4], Tobias Schorr[4], Ottmar Möhler[4], Peter Spichtinger[1], and Martina Krämer[1,3]

[1] *Institute for Atmospheric Physics, Johannes Gutenberg University, Mainz, Germany*
[2] *Zentrum für Datenverarbeitung, Johannes Gutenberg University, Mainz, Germany*
[3] *Forschungszentrum Jülich GmbH, Institute of Energy and Climate Research 7 – Stratosphere, Jülich, Germany*
[4] *Institute of Meteorology and Climate Research, Karlsruhe Institute of Technology, Karlsruhe, Germany*
[a] *now at: Deutscher Wetterdienst, Offenbach, Germany*

October 11, 2021

**Contents**

**1 General Response**

We thank both reviewers for the effort and time they spent in preparing their reviews, their helpful comments and suggestions which led to a significantly improved manuscript.

This document contains our responses to the reviewer comments together with the output of "latexdiff" to highlight the changes made to the manuscript (blue text is added, red text is removed; however note that changes made to the figures are not highlighted).

After some general responses to points raised by the reviewers, we list all individual comments (pasted to this document in blue) together with our responses (in black).

**2 Response to Reviewer #1**

**2.1 Response to General comments**

At the start of the paper, it needs to be clarified that there are two broad types of homogeneous freezing: that of solution droplet aerosols (CCN) and that of cloud-liquid/rain. It also needs to be clarified that the paper is only about the former.

We thank for this comment. Indeed, the freezing of cloud droplets which might lead to the formation of liquid origin cirrus clouds can also be homogenous ice nucleation that happens during the ascend of an air mass when passing $-38\,°C$, but in this study the focus is only on the homogeneous freezing of small and aqueous solution particles. We added some text in the introduction (see Page 2 of manuscript with changes tracked) to clarify the focus of the present study.

An intellectual weakness of the paper is that there seems to be a rather ad hoc trial of various combinations of water activity and vapour pressure schemes, with a recommendation of an optimal combination without clear arguments independently for why these schemes are best. So, the reader may infer that there is a likelihood of a balance of compensating biases which could be unsettled if a different chemical species of aerosol (not sulphate) or different thermodynamic conditions were simulated.

We do not agree with the reviewer on the point that the choice of parameterizations is an ad-hoc trial. All parameterizations are well-known in the literature and widely used in many numerical models. Moreover, each parameterization is based on laboratory measurements (hence there is empirical evidence for each) and/or on theoretical considerations (such as thermodynamic models). In essence, we collect commonly employed parameterizations from the literature and test them within the CLaMS-Ice model to identify the "best combination" that approximates the observations from the AIDA cloud chamber. We agree with the reviewer in that we cannot exclude any compensating biases that might render a particular combination of parameterizations to give the best results. In our opinion, to exclude compensating biases at all, a much more detailed understanding of the various chemical processes is needed, that ultimatively drive the freezing process. However, such an explanation is clearly out of scope of the present study, moreover, this study shows the need for more discussions on the understanding of the homogeneous freezing process. Including more simulations with different chemical species would not help to exclude compensating biases, since for each additional species new parameterizations of the physical quantities of the respective species are required and each new parameterization introduces again the question of possible compensating biases. We added a comment to the text (see Page 16 of manuscript with changes tracked) to warn the reader of possible compensating biases.

Need to argue more about whether the Nachbar scheme is more accurate than the Murphy and Koop scheme for vapour pressure. Yes, the Nachbar scheme is from recognition of a new phase of water. But what do the authors think about the likelihood of this assumption being correct ? What new information has come to light that supports this assumption ?

We thank for this comment that hints on an interesting question. However, this study is not the right place for a discussion of the accuracy of Nachbar's parameterization in comparison to Murphy and Koop's formulation (or any other) for two reasons:
(i) Such an assessment would need a detailed description of the physiochemical properties of water at cold temperatures, that is clearly beyond the scope of this work, where the influence of the water saturation formulations on the homogeneous ice nucleation is investigated. Also, to the authors knowledge, the latest degree of understanding is provided in Nachbar et al. (2019).
(ii) We think that the required details in the properties of water at cold temperatures are still not entirely understood such that a definite answer to the question is not available

at the moment, i.e. one cannot definitely confirm the correctness of the assumption in Nachbar et al. (2019).

It would be good to comment on AIDA measurements by Mangold et al. (2005) presented at EGU about lower humidities for freezing of ammonium sulphate than predicted by Koop. Was this an anomalous observation perhaps ?

The observations by Mangold et al. on the freezing properties of ammonium sulphate particles that were presented a the EGU 2005 are explained and published in Mangold et al. (2005). In essence, these particles freeze heterogeneously because a small crystal forms inside the particles during cooling that then effectively acts as an INP, hence the freezing experiments presented by Mangold et al. (2005) are not comparable to the homogeneous freezing discussed in the present study and also not to the observations described in Schneider et al. (2021).

For modellers who use the Koop scheme, we freeze CCN aerosols at temperature- and size-dependent critical saturation ratios. How should we modify our schemes ? Modellers have typically simply created a lookup table of these critical ratios from the published plot of Koop et al. (2000): Can a simple numerical fix be recommended ? If a line of best fit can be given for the "best fit" of Fig. 15a for r = 0.05 um, do we just shift the line downwards for larger sizes ?

We thank the reviewer for the interest to apply our results in the context of a larger model. However, we think that it is too early to implement these new findings in the context of a larger model, since the modified homogeneous freezing line is based merely on measurements under the laboratory conditions of the AIDA cloud chamber. To our opinion, the next (planned) step is to show that the new freezing thresholds are also valid under atmospheric conditions, for example lower vertical velocities and pressure.
We also like to note here, that Koop's homogeneous freezing line is ideally suited for most atmospheric applications. Differences to this line mainly occur at temperatures that predominantly prevail in the atmosphere in the TTL or at higher altitudes in polar winters. For these reasons, we prefer at the present stage not to provide a parameterization or look up table for the "best fit" of the new homogeneous freezing line.

**2.2 Response to Specific comments**

1. Line 24: the statement is too narrowly centered on aerosols: "In contrast, homogeneous nucleation refers to the spontaneous freezing of pre-existing solution particles". In fact, homogeneous nucleation of ice refers to spontaneous freezing of any drop, whether a solution droplet ('homogeneous aerosol freezing') or a liquid cloud- or rain-drop ('homogeneous freezing of cloud-liquid or rain').

   We included some text to clarify the focus of the study, please also see our response on the first general comment.

2. Line 363: It is interesting that, qualitatively, a similar sort of dependency on updraft speed is found with homogeneous freezing of cloud-droplets near -36 degC. With these too, the largest freeze first during ascent and an ascent-dependent fraction of all the cloud-droplets will freeze with the rest evaporating due to the ice particles lowering the humidity. There is preferential freezing of the smaller cloud- droplets. So the number of ice particles initiated increases with updraft speed. Phillips et al. (2007, JAS) parameterized this and showed that it has an order-of-magnitude impact on mesoscale averages of the ice concentration aloft.

   We thank for this comment. Indeed, the larger an aqueous particle is, the larger its probability of freezing and if enough aerosol particles are frozen, their diffusional growth lowers the humidity. However, the cooling of the air parcel due to a persistent updraft counteracts this lowering of humidity and hence influences the number of nucleated ice crystals. This mechanism is described in detail in the references Spichtinger and Gierens (2009) and Baumgartner and Spichtinger (2019), where the first of these references discusses these facts in the context of the microphysics scheme of CLaMS-Ice and the latter in a more theoretical context.

3. Line 573: The paper suggests an optimum combination: "Carslaw's method for the water activity in the nucleation routine and Luo's method in the Köhler equilibrium routine, in combination with Nachbar's formulation of the saturation vapor pressure". But what are the independent reasons for thinking that Carslaw's method is better than alternatives for water activity ? Why is Luo's method better than alternatives for the Kohler equilibrium routine ? Why is Nachbar's formulation best ? Need to provide some independent reasoning or evidence.

   Could there be a serendipitous compensation of opposing biases among these three schemes, giving the impression of realism for the wrong reasons ?

   We thank for these interesting questions. For the same reason as in our response to the third general comment, we cannot exclude the existence of compensating biases or provide independent reasons why one formulation yields superiour results in comparison to the others. These questions need a much more detailed discussion and understanding of all the chemical processes that influence the freezing process itself. In effect, the sheer existence of several different parameterizations for a physical quantity illustrates, that still not all details are fully understood. For this reason, we chose to compare the popular formulations that are also found in other studies and models.

**2.3 Response to Technical details**

1. Figure 13: this has wrong entries in the legend (lines instead of symbols) and multiple panels have identical titles and legends without clarity from the figure caption.

   We updated the figures to add the missing symbols to the respective legends. In addition, we modified several captions to more clearly indicate the meaning of the various symbols, lines and colors.

**3 Response to Reviewer #2**

**3.1 Response to Major concern**

In a few simulations, the aerosol equilibrium and nucleation rate calculations employ different, inconsistent formulas for water activity within the aqueous aerosol. This results in an inconsistency, which is acknowledged for example p 29 line 515. One of the inconsistent numerical experiments shows an improved agreement with AIDA ice onset observations (Fig. 15).

In my opinion, this inconsistency alters the original concept of Koop et al. (2000) which not only synthesizes earlier laboratory experiments but also aims at simplicity, taking advantage of thermodynamic constraints. Those include:

$$a_w^i = \frac{p_{i,sat}}{p_{sat}} \tag{1}$$

and Köhler theory for aqueous aerosols in equilibrium, namely:

$$a_w = S_i a_w^i \exp\left(\frac{-2\sigma}{R_v T \rho r}\right) = RH_w \exp\left(\frac{-2\sigma}{R_v T \rho r}\right) \tag{2}$$

In Equation 2, the resulting equilibrated water activity in the aerosol, $a_w$, depends on the analytical expression of water activity in the solution only through the factor $r$. If the curvature is neglected (i.e. if $r$ is sufficiently large), Equation 2 reduces to the equilibrium formula proposed by Koop et al. (2000):

$$a_w = S_i a_w^i \tag{3}$$

in which aerosol water activity is set by environmental conditions only and is virtually independent of aerosol properties. As a consequence, the nucleation probability does not depend (or only marginally, through $r$) on the formula used for water activity in the solute. This is actually noticed (though not explained) by the authors: in consistent set-ups, they obtain similar nucleation thresholds although the formulas used for water activity are different (p 29 lines 518-519).

However, by employing inconsistent expressions for water activity between the aerosol equilibration (Köhler) and the freezing rate calculation, the authors artificially break the link between environmental conditions and the nucleation rate implied in Koop et al. (2000). Namely, instead of having the activity in the solute given by Eq. 3 (as an approximation to Eq. 2), it now follows :

$$a_w = f_{1\to2}(S_i a_w^i, T) \tag{4}$$

where the function $f_{1\to2}$ converts activity from formulation 1 ($a_{w,1}$ used in the Koehler routine) to formulation 2 ($a_{w,2}$ used in the nucleation rate computation) and is defined such that:

$$a_{w,2}(x(r), T) = f_{1\to2}(a_{w,1}(x(r), T), T) \tag{5}$$

I believe this implicit replacement of Eq. 2 (or 3) by Eq. 4 (or a similar one with Kelvin correction) is the reason for the significant sensitivity of the nucleation thresholds to water activity formulations in the authors' simulations, not only in Sect. 4.3 but also in Sect. 4.4. This important point needs to be accounted for and discussed early on by the authors. For clarity, I also would also ask them to drop the "most" when referring to the "most consistent" configurations, since they are "just" consistent. Note that this comment does not affect the conclusions regarding the impact of the formula used for water saturation.

We thank the reviewer for this excellent presentation of the argument. We fully agree and added some text to the manuscript along the lines of the Reviewer's presentation (see Page 16 of the manuscript with changes tracked). Moreover we followed the suggestion and dropped the word "most".

**3.2 Response to Other comments**

- I noticed differences between figures 6 and 8 at low vertical velocity where the small time step result exceeds that obtained with the large time step by a factor of about 2. This should be explained or corrected.

  We thank for this observation. Indeed, there are slight deviations in the number of nucleated ice crystals at low vertical velocities with the small timesteps. These deviations do occur in the CLaMS-Ice simulations but seem to be not present in the simulations presented in Spichtinger and Gierens (2009), where the employed microphysics scheme is described. We assume these deviations to be the result of some numerical problem that becomes significant for small timesteps only. We will definitely need to investigate the reason for the deviation but since the deviation is comparatively small and the model usually runs with larger timesteps, we decided to leave the simulations as they are for now.

- Related to my main comment, I disagree with the authors regarding the impact of aerosol properties (in particular, aerosol radius $r$). At the beginning of the paper, it is argued (as in Koop et al., 2000; Kärcher and Lohmann, 2002) that the dependency on aerosol size is moderate, and cannot explain the differences between the AIDA chamber experiments and simulations based on the classical Koop approach. Comparing Fig. 10 with Fig. 11 demonstrates that the impact of the aerosol size distribution, albeit present, remains limited.
  Then, on several instances, the authors attribute changes in the nucleation threshold to changes in the aerosol radius $r$ associated with Köhler equilibrium: Sect. 2.2, lines 446-460 (which seemingly contradict lines 446-460 where the primary importance of ice activity formulation is recognized) and Sect. 2.4, lines 526-546. As decribed in my main comment, I would guess that the apparent sensitivity to aerosol size is an artifact due to the inconsistent use of different solute water activity formulations.

  We thank for this comment which shows us that some more explanations should be given in this context: The theory of homogeneous freezing at the beginning of our manuscript mainly describes the homogeneous freezing of a single aerosol particle, i.e. the theory provides the formula $P = 1 - \exp(-JVt)$ for the freezing probability of a particle with volume $V$. Consequently, the same freezing behaviour is expected for a population of equally sized particles, i.e. a monodisperse size distribution. However, in our CLaMS-Ice simulations we do not have a monodisperse size distribution but a lognormal size distribution with parameter $\sigma_a$ that determines the width of the distribution. If $\sigma_a$ is close to unity, most aerosol particles have comparable sizes and the freezing behaviour is close to the case of a monodisperse distribution. If $\sigma_a$ is larger (as $\sigma_a = 1.6$ in the simulation results shown), the size distribution is much broader and significantly larger aerosol particles are present. Consequently the geometric mean radius $r_a$ becomes less expressive for the broader aerosol population. In this case, the freezing is determined by the freezing of the large particles and the freezing starts much earlier than one would expect if only the value of $r_a$ is considered. In effect, a broad size distribution features a sensitivity to the aerosol size in the sense, that the freezing is determined by the large particles. We included some sentences to hint the reader on this fact (see Page 12 in the manuscript with changes tracked) and also included an appendix (Appendix C; see Page 38 in the manuscript with changes tracked) that shows a narrow and a broad (normalized) aerosol size distribution for illustration. The connection to the Köhler equilibrium comes from the fact, that the dry aerosol size distribution is converted to a size distribution for the aqueous aerosol population at model ambient conditions (by the Köhler routine) and the freezing is subsequently computed. Consequently, a broad dry aerosol size distribution gets transformed into a broad aqueous aerosol size distribution where the large aerosol particles determine the freezing.

- The paper rests solely on numerical investigations, whereas the authors have made valuable contributions to the theoretical analysis of homogeneous freezing of aqueous aerosols (e.g. Baumgartner and Spichtinger, 2019). Couldn't some of this analysis be useful here ? It would help explain and synthesize the results (such as the insensitivity of nucleated ice crystal number to the activity formulation).

We thank for this interesting suggestion which we discussed internally. The theoretical analysis in Baumgartner and Spichtinger (2019) was only possible because we simplified the governing equations (e.g. we eliminated the equation that describes the evolution of the ice crystal mass). For this reason, the effects that are discussed in the present study are no longer represented in the simplified equations from Baumgartner and Spichtinger (2019). However, it would be an interesting avenue for future research to explore the possibilities of using the same theoretical methods to the current more detailed aspects.

- Why are the freezing onset simulations limited to the range $190 - 230\,\mathrm{K}$? The relevant temperature range for cirrus extends down to $\sim 180 - 185\,\mathrm{K}$ (tropical tropopause layer) as do the experiments of Schneider et al. (2021).

We thank for this question whose answer has two aspects. First, the choice of temperatures was inspired by the study of Kärcher and Lohmann (2002), who chose the three temperatures $196\,\mathrm{K}$, $216\,\mathrm{K}$, and $236\,\mathrm{K}$ which was also adopted in the study of Spichtinger and Gierens (2009) where the two-moment microphysics scheme is described.
The second aspect is that our study and the experiments described in Schneider et al. (2021) were done in parallel and, since measurements at the AIDA chamber are very challenging at such cold temperatures, successful experiments in this temperature range could be only achieved at the latest stage of the joined project when the modelling part was already completed. The reason that the simulations were not expanded to $185\,\mathrm{K}$ was simply that the main author switched to another research area, as can be seen from his new affiliation. However, we are convinced that this extension would have been a nice addition to the study, but would not have changed the main results.

- The goal of the study is not entirely clear, is it (a) an evaluation paper for the nucleation scheme of CLaMS-ice, or (b) an investigation of the impact of water activity and saturation formulation on homogeneous freezing calculations (as suggested by the title and introduction)? If (b), I would encourage the authors to rephrase part of the text to emphasize that most conclusions apply in general and not only to their specific numerical model. They could remove/shorten some of the many references to the code (l 324, ...).

We thank for this comment. Indeed, the paper is not intended as an evaluation paper of CLaMS-Ice but as a study that exemplifies the impact of the parameterizations on the homogeneous nucleation at the example of the CLaMS-Ice model. We discussed about removing the references to the code but decided to leave them in the text since some of the arguments do rely on the code itself, e.g. the different places in the code where we can change the formulation of the water activity.

- The paper could be more concise. For instance, some figures and discussions could be moved to the appendix/ supplementary information. I am in particular referring to Fig 8 (p 18-19) which shows the sensitivity of the model results to the time step and illustrates that, for large vertical velocities, the calculations in Figs 6 and 7 had not converged. I would discuss this in an appendix and only keep figures 6 and 7 with the converged (i.e. 5 ms) time step. Also, some panels in Figs. 6-9 could be moved to the supplementary information, to limit the number of panels in the main body of the paper and make the relevant information more accessible.

Thanks a lot for this comment, which seemingly needed a reader with a certain distance to the study. We fully agree on the missing degree of conciseness and took the comment as a motivation to move some parts of the text into the appendix (e.g. the illustration of the results with the small timestep, as suggested, see Page 38 of the manuscript with changes tracked). However, in our opinion moving individual panels from the figures of the main text to an appendix or supplementary material could result in confusion due to the even more scattered informations, hence we prefer to keep the figures with all panels in one place.

- It would be useful to have a table or schematic summarizing where each parameterization/ formula is used in the model for a given set of experiments (i.e. the water vapor saturation and water activity parameterizations).

We added a table (Table 2, see page 17 of the manuscript with changes tracked) to summarize the key aspects of the various experiments that are shown in the main text.

**3.3 Response to Specific comments**

1. p 3 l 60: remove brackets

   Done.

2. p 3 l 67: is this comment relevant in the introduction ? the point about latent heat release is not discussed further in the text and does not seem essential to me.

   Indeed, the latent heat release is not further discussed in the text, however we prefer to keep this point for two reasons: (i) Including the comment avoids any ambiguity and (ii) from our experience with the simulations of homogeneous nucleation events we observed that the latent heat release actually influences the number of ice crystals at warmer temperatures, e.g. from 230 K onward, such that the number of nucleated ice crystals might change up to a factor of roughly two.

3. p 15 Sect 3.4: I would specify already here the aerosol characteristics and water saturation formula used by Kärcher and Lohmann (2002)

   We followed this suggestion and included these informations in Section 3.4.

4. p 24 l 445: the Spichtinger et al. paper is not yet published. Putting a version on archive would be useful.

   Indeed, this paper is not yet published und we currently work on the manuscript. We will follow the Reviewer's suggestion and put a version on arXiv or submit it to ACPD in due time.

5. Fig 10, 11: the equation number of Koop activity is missing in the legend

   The intention of the bracket "(Eq.)" was to remind the reader that the Koop-formulation for $a_w$ is based on the equilibrium approach. We thank for pointing us on the possible confusion with the word "Equation". In order to avoid any ambiguity, we adapted the legend and eliminated the bracket "(Eq.)".

6. p 29 l 518-519: This is expected I believe (see main comment).

   Yes; we added some words (see Page 29 of the manuscript with changes tracked) to remind the reader why this is to be expected.

7. p 29, Eq (21): The reference formula/value used for the surface tension $\sigma$ should be specified.

   The parameterization of the surface tension was taken from Tabazadeh et al. (2000),

which is based on the work by Myhre et al. (1998). We included the reference in the text. In addition, we also included the reference for the density.

8. p 32, lines 582-583: the authors can reproduce the nucleated number concentration "as long as the mean size and width of the size distribution of the aqueous solution particles from the AIDA experiments are known". How do they fit the dry aerosol radius to match the hydrated observed radius ? How strongly do the presented results depend on this fit? Furthermore, is the distribution of dry aerosols in the previous figures consistent with the aqueous aerosol in the AIDA experiments shown in Fig. 2 of Schneider et al. (2021) ($r \sim 250\,\text{nm}$) ?

The exact fitting process of the observed size distributions is described in detail in Schneider et al. (2021). In essence, the aerosol size distribution is measured outside of the cloud chamber and necessarily equals already the hydrated aerosol particles. After the measurement, a lognormal size distribution is fitted to the observed size distribution, where the focus of the fitting process is on the large particle mode because these particles can be measured with lower uncertainties. The parameters of the fitted aerosol size distribution is then indicative of the dry aerosol size distribution. Arguably, the values of the parameters of the dry size distribution might be slightly different, but the general effect of these differences (smaller or larger values for $r_a$ and/or $\sigma_a$) is discussed in our manuscript.

9. Fig. 15: If possible, error bars should be provided for the experimental values.

We thank for this comment and updated Figure 14 (which was Figure 15 in the initially submitted manuscript) to now include the most recent corrected experimental data values together with error bars. As can be easily seen, the ice crystal number concentrations from the CLaMS-Ice model is roughly within an interval of factor two to the experimental values, what is deemed a good agreement.

**References**

Baumgartner, M. and Spichtinger, P. (2019). "Homogeneous nucleation from an asymptotic point of view". In: *Theoretical and Computational Fluid Dynamics* 33.1, pp. 83–106.

Kärcher, B. and Lohmann, U. (2002). "A parameterization of cirrus cloud formation: Homogeneous freezing of supercooled aerosols". In: *Journal of Geophysical Research: Atmospheres* 107.D2.

Koop, T., Luo, B., Tsias, A., and Peter, T. (2000). "Water activity as the determinant for homogeneous ice nucleation in aqueous solutions". In: *Nature* 406, pp. 611–614.

Mangold, A., Wagner, R., Saathoff, H., Schurath, U., Giesemann, C., Ebert, V., Krämer, M. ., and Möhler, O. (2005). "Experimental investigation of ice nucleation by different types of aerosols in the aerosol chamber AIDA: implications to microphysics of cirrus clouds". In: *Meteorologische Zeitschrift* 14.4, pp. 485–497.

Myhre, C. E. L., Nielsen, C. J., and Saastad, O. W. (1998). "Density and Surface Tension of Aqueous H2SO4 at Low Temperature". In: *Journal of Chemical & Engineering Data* 43.4, pp. 617–622.

Nachbar, M., Duft, D., and Leisner, T. (2019). "The vapor pressure of liquid and solid water phases at conditions relevant to the atmosphere". In: *The Journal of Chemical Physics* 151.6, p. 064504.

Schneider, J., Höhler, K., Wagner, R., Saathoff, H., Schnaiter, M., Schorr, T., Steinke, I., Benz, S., Baumgartner, M., Rolf, C., Krämer, M., Leisner, T., and Möhler, O. (2021). "High Homogeneous Freezing Onsets of Sulfuric Acid Aerosol at Cirrus Temperatures". In: *Atmospheric Chemistry and Physics Discussions*. In preparation.

Spichtinger, P. and Gierens, K. M. (2009). "Modelling of cirrus clouds – Part 1a: Model description and validation". In: *Atmospheric Chemistry and Physics* 9.2, pp. 685–706.

Tabazadeh, A., Martin, S. T., and Lin, J.-S. (2000). "The effect of particle size and nitric acid uptake on the homogeneous freezing of aqueous sulfuric acid particles". In: *Geophysical Research Letters* 27.8, pp. 1111–1114.